# Topological prethermal strong zero modes on superconducting processors

Feitong Jin[1,11], Si Jiang[2,3,11], Xuhao Zhu[1,11], Zehang Bao[1], Fanhao Shen[1], Ke Wang[1], Zitian Zhu[1], Shibo Xu[1], Zixuan Song[1], Jiachen Chen[1], Ziqi Tan[1], Yaozu Wu[1], Chuanyu Zhang[1], Yu Gao[1], Ning Wang[1], Yiren Zou[1], Aosai Zhang[1], Tingting Li[1], Jiarun Zhong[1], Zhengyi Cui[1], Yihang Han[1], Yiyang He[1], Han Wang[1], Jia-Nan Yang[1], Yanzhe Wang[1], Jiayuan Shen[1], Gongyu Liu[1], Jinfeng Deng[1], Hang Dong[1], Pengfei Zhang[1], Weikang Li[2,4], Dong Yuan[2], Zhide Lu[3], Zheng-Zhi Sun[2,5], Hekang Li[1], Junxiang Zhang[1], Chao Song[1], Zhen Wang[1,5], Qiujiang Guo[1,5 ✉], Francisco Machado[6,7], Jack Kemp[7,8], Thomas Iadecola[9,10], Norman Y. Yao[7], H. Wang[1,5 ✉] & Dong-Ling Deng[2,3,5 ✉]

Symmetry-protected topological phases[1-4] cannot be described by any local order parameter and are beyond the conventional symmetry-breaking model[5]. They are characterized by topological boundary modes that remain stable under symmetry respecting perturbations[1-4,6-8]. In clean, gapped systems without disorder, the stability of these edge modes is restricted to the zero-temperature manifold; at finite temperatures, interactions with mobile thermal excitations lead to their decay[9-11]. Here we report the observation of a distinct type of topological edge mode[12-14], which is protected by emergent symmetries and persists across the entire spectrum, in an array of 100 programmable superconducting qubits. Through digital quantum simulation of a one-dimensional disorder-free stabilizer Hamiltonian, we observe robust long-lived topological edge modes over up to 30 cycles for a wide range of initial states. We show that the interaction between these edge modes and bulk excitations can be suppressed by dimerizing the stabilizer strength, leading to an emergent U(1) × U(1) symmetry in the prethermal regime of the system. Furthermore, we exploit these topological edge modes as logical qubits and prepare a logical Bell state, which exhibits persistent coherence, despite the system being disorder-free and at finite temperature. Our results establish a viable digital simulation approach[15-18] to experimentally study topological matter at finite temperature and demonstrate a potential route to construct long-lived, robust boundary qubits in disorder-free systems.

Symmetry and topology are fundamental to characterizing quantum phases of matter[1,2]. Their interplay gives rise to a rich variety of exotic phases[1-4] that cannot be described by the traditional Landau–Ginzburg symmetry-breaking model[5]. A prominent example is symmetry-protected topological (SPT) phases, which feature nonlocal order parameters and topological boundary modes that are robust against local perturbations respecting the protected symmetry[1-4,6-8]. These robust boundary modes provide an opportunity to store and process quantum information in a perturbation-resilient fashion[19]. In a clean, gapped system without disorder, these edge modes are typically restricted to the zero-temperature manifold[9-11]. At finite temperature, these edge modes would interact strongly with thermal excitations in the bulk and decohere rapidly. Realizing robust topological edge modes at finite temperatures is crucial to understanding hot SPT phases of matter and has potential applications in building a noise-resilient quantum memory[9].

A popular strategy to stabilize topological edge modes at finite temperature involves adding strong disorder so as to make the system many-body localized[20-22]. In such a scenario, bulk thermal excitations become localized, preventing them from scattering with and decohering the topological edge modes[23-25]. Despite exciting progress along this direction[26-28], the stability of many-body localization is still under active debate[29-32], which limits our understanding of the long-time behaviour of localization-based SPT phases at finite temperature. Moreover, the presence of strong disorder slows down equilibration, making it difficult to unambiguously distinguish genuine late-time dynamics from early-time transient behaviours in experiments[26-28]. An alternative strategy is to suppress the interactions between bulk excitations and edge modes by emergent symmetries, rather than localization[12-14]. In this case, the system can be disorder-free and bulk excitations remain mobile, but the additional symmetry constraints give rise to approximately conserved edge states that remain, effectively, decoupled from

[1]School of Physics, ZJU-Hangzhou Global Scientific and Technological Innovation Center, and Zhejiang Key Laboratory of Micro-nano Quantum Chips and Quantum Control, Zhejiang University, Hangzhou, China. [2]Center for Quantum Information, IIIS, Tsinghua University, Beijing, China. [3]Shanghai Qi Zhi Institute, Shanghai, China. [4]Instituut-Lorentz, Universiteit Leiden, Leiden, The Netherlands. [5]Hefei National Laboratory, Hefei, China. [6]ITAMP, Harvard-Smithsonian Center for Astrophysics, Cambridge, MA, USA. [7]Department of Physics, Harvard University, Cambridge, MA, USA. [8]TCM Group, Cavendish Laboratory, Ray Dolby Centre, University of Cambridge, Cambridge, UK. [9]Department of Physics and Astronomy, Iowa State University, Ames, IA, USA. [10]Ames National Laboratory, Ames, IA, USA. [11]These authors contributed equally: Feitong Jin, Si Jiang, Xuhao Zhu. ✉e-mail: qguo@zju.edu.cn; hhwang@zju.edu.cn; dldeng@tsinghua.edu.cn

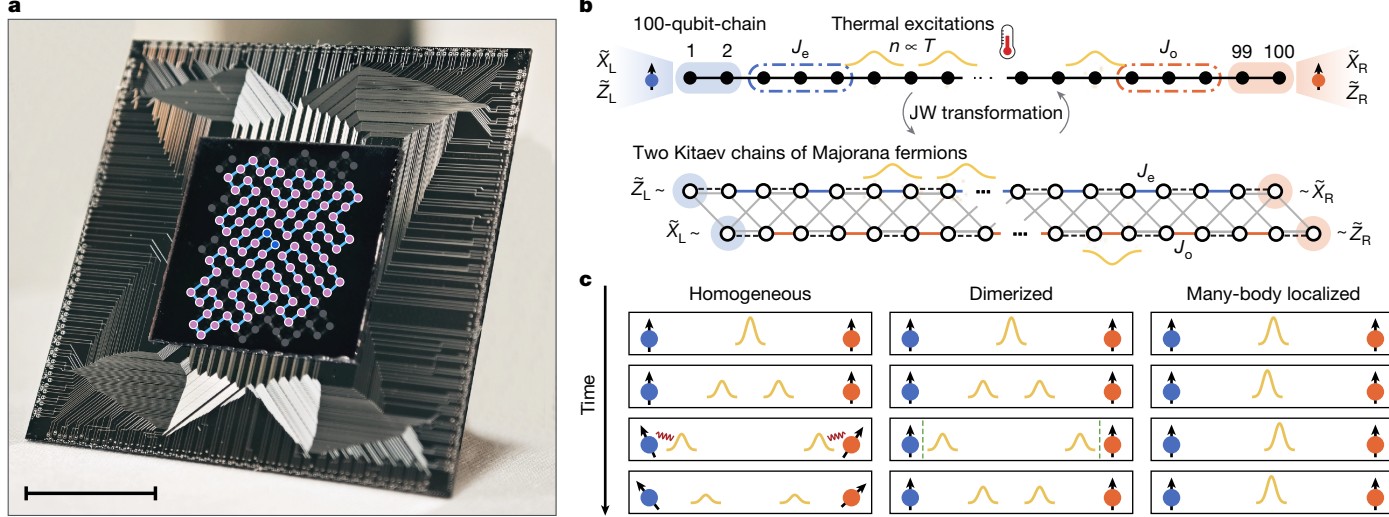

**Fig. 1 | The 125-qubit quantum processor and the theoretical model.**
**a**, Photograph of the superconducting quantum processor. The 100 qubits used to construct the 1D chain are highlighted with circles, with two edge qubits marked in dark blue and the other qubits in pink. The couplers actively used are highlighted with light blue lines. **b**, Schematic of the 1D Hamiltonian in equation (1) and its representation in the Majorana fermion picture. Three-body stabilizers $\{\sigma_{i-1}^z \sigma_i^x \sigma_{i+1}^z\}$ at even and odd sites, shown as blue and orange dashed frames, can have different strengths denoted by $J_e$ and $J_o$, respectively. Two spin-1/2 edge modes are situated at the two ends of the chain, characterized by $\tilde{Z}_L, \tilde{X}_L$ for the left edge and $\tilde{Z}_R, \tilde{X}_R$ for the right edge. At finite temperatures, thermal excitations (yellow wave packets) emerge in the bulk, flipping the values of stabilizers. After the Jordan–Wigner (JW) transformation, the 1D qubit chain is mapped into two Kitaev chains, in which the upper chain inherits

the even-site interaction strength $J_e$ (blue lines) and the lower chain inherits the odd-site interaction strength $J_o$ (orange lines). Two edge modes are transformed into four Majorana fermions at the ends of two chains. Single-qubit $\sigma_i^x$ terms (black dashed lines) become couplings of onsite Majorana pairs, and two-qubit $\sigma_i^x \sigma_{i+1}^x$ interactions (grey lines) bridge the two chains. **c**, Schematic of thermal excitation dynamics and their interactions with edge modes. Thermal excitations (yellow wave packets) can propagate through the chain under perturbations. In the homogeneous regime (left, $J_o = J_e$), edge–bulk interactions at the boundaries decohere and ruin the edge modes. Whereas in the (off-resonant) dimerized regime (middle, $J_o \neq J_e$), these interactions are markedly suppressed, resulting in long-lived robust edge modes at up to infinite temperature. In the many-body localized scenario (right), transport is forbidden and thermal excitations remain localized without influencing the boundaries. Scale bar, 10 mm (**a**).

the bulk. These topological edge states form so-called prethermal strong zero modes, which feature nearly exponentially long coherence times even at infinite temperature[12–14,33–36]. Pioneering experiments have observed signatures of topological edge modes at up to infinite temperature in periodically driven systems with strong disorder[37–39]. Yet, the observation of long-lived finite-temperature topological edge modes protected in disorder-free systems remains a notable challenge and has evaded experiments so far.

Here we report such an observation with a newly developed high-performance 125-qubit superconducting quantum processor (Fig. 1). We select 100 neighbouring qubits arranged in a one-dimensional (1D) chain (Fig. 1a), featuring median fidelities of simultaneous single- and two-qubit gates of about 0.9995 and 0.995, respectively. This enables us to successfully implement the dynamics of a prototypical SPT Hamiltonian (Fig. 1b) in different regimes. We prepare the system in different initial states with different energies, which correspond to different effective temperatures, and then evolve it under the SPT Hamiltonian with varying parameters. We observe that, in the presence of thermal excitations, the lifetime of edge states is greatly enhanced in the dimerized regime with spatially periodically modulated couplings, in stark contrast to the fast decay in the homogeneous case. This distinction also manifests in the spatial profiles of edge modes, which become more localized as the couplings deviate from the homogeneous regime. To reveal the underlying mechanism, we measure the site-resolved dynamics of mobile excitations. Although the thermal excitations are mobile, an approximate U(1) × U(1) symmetry emerges in the dimerized case that suppresses the bulk–edge interactions. This stands in sharp contrast to the many-body localized scenario in which the interactions are suppressed because of the localization of bulk excitations (Fig. 1c). We further confirm this prethermal suppression mechanism by measuring the energy spectrum, in which an extra gap gradually opens as the chain dimerizes, explaining the origin of the emergent

symmetry. Furthermore, we prepare a logical Bell state encoded within these topological edge modes and demonstrate its substantially prolonged coherence time at finite temperature in the dimerized and off-resonant regime. This shows that the edge modes have potential applications towards building a noise-resilient finite-temperature quantum memory.

## SPT Hamiltonian and its implementation

We consider a 1D Hamiltonian with an even number of qubits denoted by $N$ (Fig. 1b):

$$H = H_0 + H_1,$$
$$H_0 = J_e \sum_{i=1}^{\frac{N}{2}-1} \sigma_{2i-1}^z \sigma_{2i}^x \sigma_{2i+1}^z + J_o \sum_{i=1}^{\frac{N}{2}-1} \sigma_{2i}^z \sigma_{2i+1}^x \sigma_{2i+2}^z,$$
$$H_1 = h_x \sum_{i=1}^{N} \sigma_i^x + V_{xx} \sum_{i=1}^{N-1} \sigma_i^x \sigma_{i+1}^x,$$

(1)

where $\hbar$ is set to 1, $\sigma_i^{x,z}$ are Pauli operators acting on the $i$th qubit, $J_e$ denotes the strength of three-body stabilizer terms centred around even sites, $J_o$ denotes the strength of three-body stabilizer terms centred around odd sites and $h_x$ and $V_{xx}$ are parameters characterizing the transverse field and interaction strength, respectively. In the limit of $h_x, V_{xx} \to 0, H = H_0$ and its eigenstates are the 1D cluster stabilizer eigenstates[40]. The two manifolds at the bottom and top of the spectrum, in which the expectation values of stabilizers $\{\sigma_{i-1}^z \sigma_i^x \sigma_{i+1}^z\}$ all equal to −1 or +1, both correspond to zero temperature. In our experiments, we choose the states with all stabilizers equal to +1 in the top manifold (denoted as $\{|\Psi_0\rangle\}$) as the zero-temperature states. Within $\{|\Psi_0\rangle\}$, the degeneracy is fourfold, hosting two nontrivial spin-1/2 topological

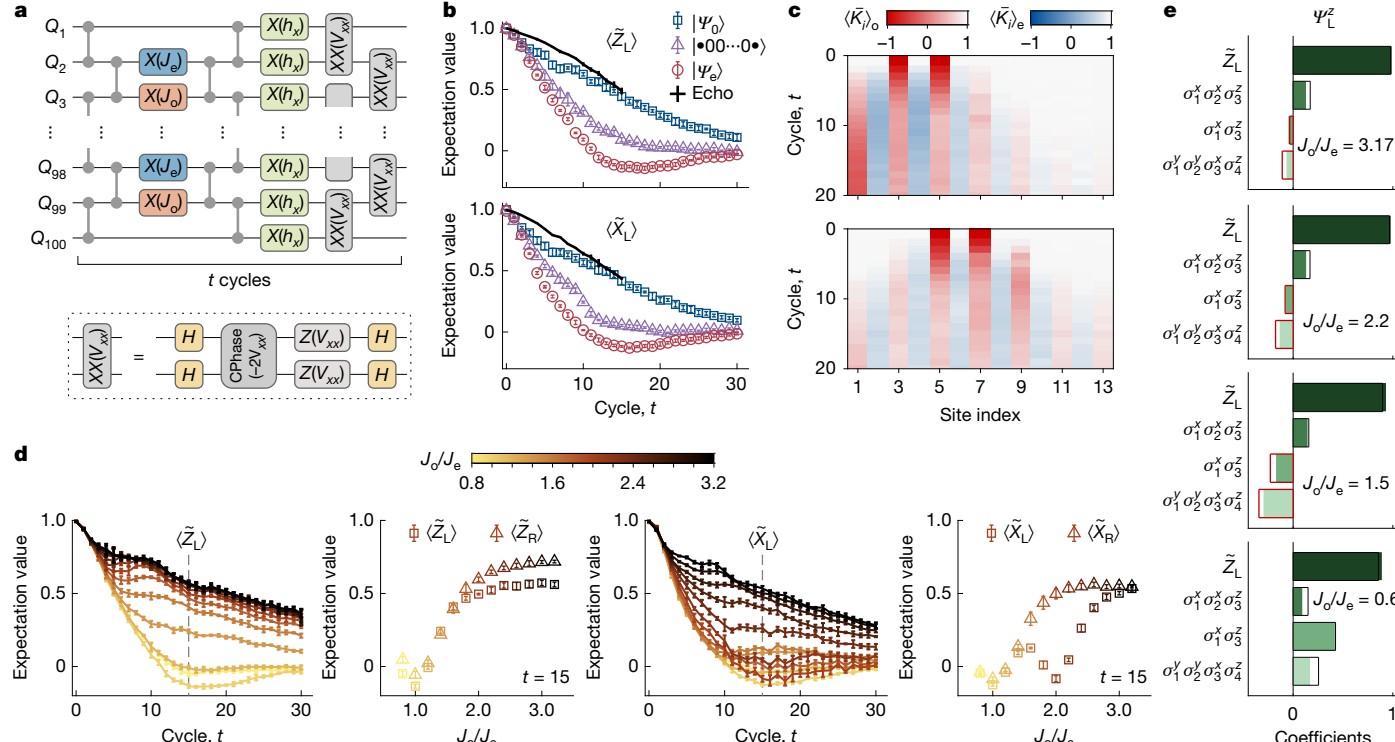

**Fig. 2 | Robust topological edge modes at up to infinite temperature.**
**a**, Quantum circuit for implementing $U(\delta t)$, which emulates a single-step
evolution (288 ns) under the Hamiltonian in equation (1). The system is initialized
in either the manifold $\{|\Psi_0\rangle\}$ (excitation number $n = 0$, corresponding to zero
temperature), the manifold $\{|\Psi_e\rangle\}$ ($n \neq 0$, finite temperature), or the product
states $|\cdot00...0\cdot\rangle$ (effectively infinite temperature), and evolved with $U(\delta t)$ for $t$
cycles. Here, $J_o$, $J_e$ and $h_x$ are parameterized into the rotation angle $\theta$ around the
$x$ axis of the Bloch sphere [$X(\theta)$]. $V_{xx}$ is encoded in a combination of controlled-
phase gates [CPhase($-2V_{xx}$)] and $Z$ phase gates [$Z(V_{xx})$]. **b**, Measured time
dynamics for the left edge operators in the homogeneous case ($J_o = J_e = \pi/5$).
Black lines show the results of echo circuits, which estimate the decay
caused by circuit errors (see Supplementary Information section 2G for more
data and discussions). **c**, Measured site-resolved dynamics of normalized

expectation value $\overline{\langle K_i \rangle}$ for bulk stabilizers $\{\sigma_{i-1}^z \sigma_i^x \sigma_{i+1}^z\}$ and edge operator $\widetilde{X}_L$ in the
homogeneous case ($J_o = J_e = \pi/5$) near the left edge. The nearest excitations to
the left edge are initialized at $\{Q_3, Q_5\}$ (top) and $\{Q_5, Q_7\}$ (bottom). **d**, Measured
time dynamics of the left edge operators with fixed $J_e = \pi/5$ and varying $J_o$.
Resonant processes lead to enhanced decay rates at $J_o/J_e = 1$ for $\widetilde{Z}_L$ and $J_o/J_e = 1, 2$
for $\widetilde{X}_L$. Error bars in **b** and **d** represent the standard deviation over five rounds
of measurements, with each taking 10,000 shots. The time dynamics of the
right-edge operators are shown in Extended Data Fig. 3. **e**, Spatial profile of
the prethermal strong zero mode $\Psi_L^z$. The solid boxes denote theoretical
predictions, with black frames highlighting the positive values and red frames
highlighting the negative values. The coefficients are obtained by averaging
the late-time dynamics over cycles from $t = 25$–40, with the sum of their squares
normalized to unity.

edge modes protected by a $\mathbb{Z}_2 \times \mathbb{Z}_2$ symmetry, where each $\mathbb{Z}_2$ is gener-
ated by the products of $\sigma_i^x$ over even or odd sites. These SPT edge modes
are characterized by logical operators $\overline{X}_L = \sigma_1^x \sigma_2^z$, $\overline{Z}_L = \sigma_1^z$ for the left
edge and $\overline{X}_R = \sigma_{N-1}^z \sigma_N^x$, $\overline{Z}_R = \sigma_N^z$ for the right edge (Fig. 1b and Supple-
mentary Information section 1C). In the presence of interactions $H_1$,
the edge modes hybridize, leading to a finite lifetime that scales expo-
nentially with the system size $N$. As the temperature increases, the
system approaches the centre of the spectrum, occupying more
excited states in which some of the stabilizers are flipped. The interac-
tions $H_1$ allow these excitations to propagate through the system, reach
the boundaries and decohere the edge states, resulting in a notably
shorter lifetime than the hybridization time (Supplementary Informa-
tion section 1A).

We emulate many-body dynamics under the Hamiltonian in equa-
tion (1) with $N = 100$ superconducting qubits using first-order Trotter
decomposition $U(\delta t) = U_1(\delta t)U_0(\delta t)$, where $U_1(\delta t) = e^{-iH_1\delta t}$ and
$U_0(\delta t) = e^{-iH_0\delta t}$. Implementing $U(\delta t)$ is challenging because three-body
interactions do not arise naturally in superconducting platforms, lead-
ing to large circuit depths. As shown in Fig. 2a, even a single time step
$U(\delta t)$ demands a deep circuit with six layers of two-qubit gates and
three layers of single-qubit gates, corresponding to a 288-ns running
time (Supplementary Information sections 2B and 2C). Therefore, the
high performance of the quantum processor (Supplementary Informa-
tion section 2A) is crucial for observing coherent dynamics under $U$

before the accumulated experimental errors dominate. In our experi-
ments, we achieve low-error quantum gates at the 100-qubit scale, with
median simultaneous single- and two-qubit gate fidelities of about
0.9995 and 0.995, respectively (Extended Data Fig. 1). We set $\delta t = 0.5$,
$J_e = \pi/5$, $h_x = 0.11$ and $V_{xx} = 0.2$ and tune the odd-site stabilizer strength
$J_o$ to observe distinct dynamical regimes. We note that the heating
induced by Trotterization errors is suppressed within our experimen-
tal timescale because of Floquet prethermalization[41–43] (Methods and
Supplementary Information section 1B).

## Robust edge modes at infinite temperature

We first explore the influence of bulk excitations on edge modes in the
homogeneous regime ($J_e = J_o$). We start by contrasting the experimen-
tally measured time dependence of the edge modes when the system
is initialized in the manifold $\{|\Psi_0\rangle\}$ versus product states $|\cdot00...0\cdot\rangle$ in
Fig. 2b (see Methods and Extended Data Fig. 2 for initial state prepara-
tion). The latter, manifesting as an effectively infinite-temperature
state with poorly protected edge modes, decays much faster. Although
$\{|\Psi_0\rangle\}$ is not the exact zero-temperature state of the system in the pres-
ence of an interaction term $H_1$, it resides in the low-temperature regime,
leading to limited effects of excitations on the edge modes. As such,
the observed decay is attributed to external experimental imperfec-
tions, especially circuit errors. This is verified by the agreement

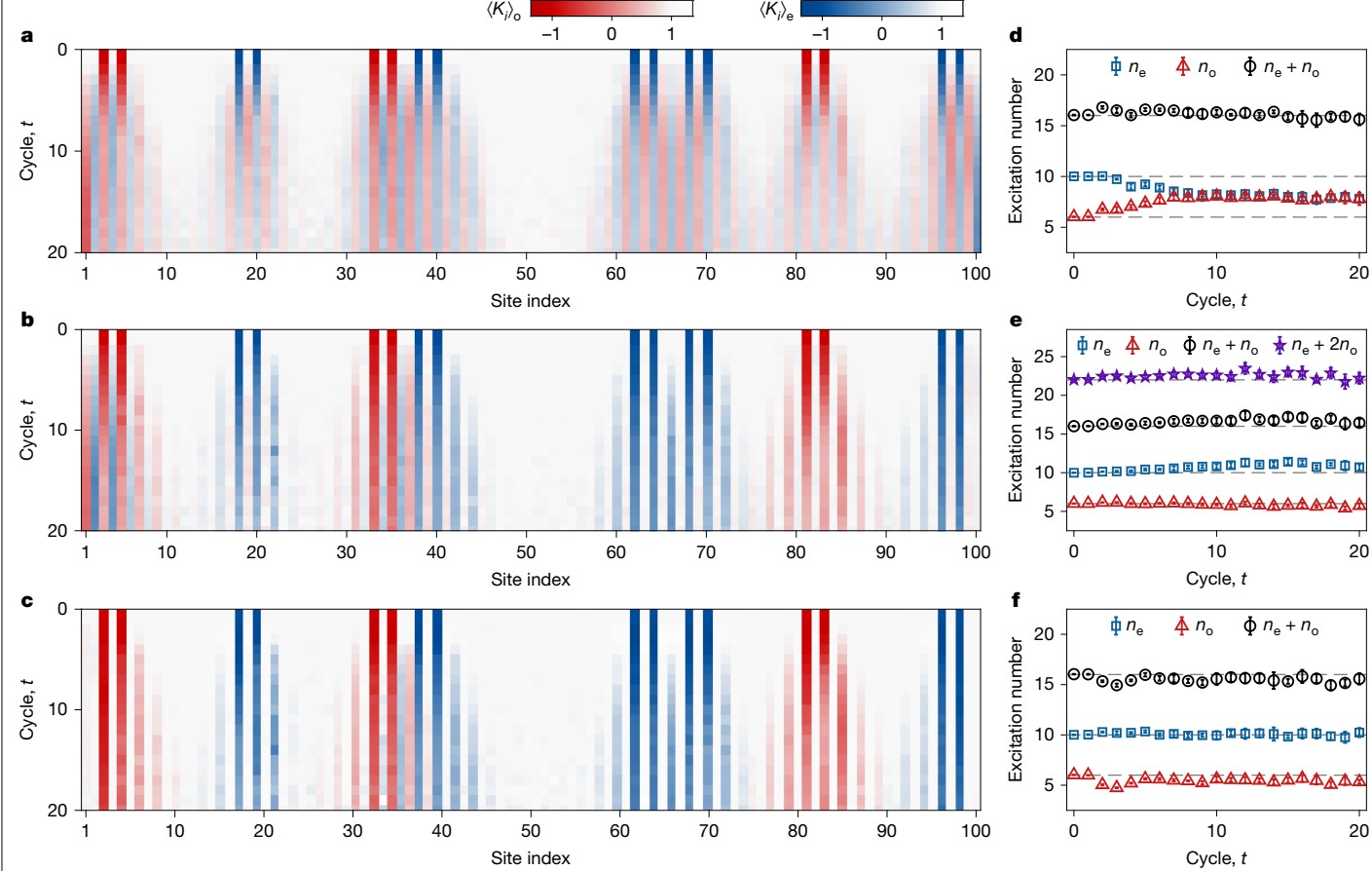

**Fig. 3 | Excitation dynamics and the emergent U(1) × U(1) symmetry.**
**a**–**c**, Measured site-resolved dynamics of normalized expectation value $\overline{\langle K_i \rangle}$ for the homogeneous ($J_o = J_e = \pi/5$) (**a**), the dimerized but resonant ($J_o = 2J_e = 2\pi/5$) (**b**) and the dimerized and off-resonant ($J_o = 3.17J_e = 3.17\pi/5$) (**c**) cases. For $\overline{\langle K_i \rangle}$ at odd (even) sites, colour bars are chosen to be red (blue) for a better visualization of the excitation dynamics. **d**–**f**, Measured time dynamics of the total excitation number $n$, and of the excitation number at even ($n_e$) and odd ($n_o$) sites, which are extracted from **a** to **c**. In the homogeneous case (**d**), the values of $n_e$ and $n_o$ gradually converge, yet their sum remains approximately constant, reflecting

the U(1) symmetry on the total excitation number $n$ in the bulk. In the dimerized but resonant case (**e**), the exchange of excitations between two Kitaev chains, which happens near the left edge, can be observed through the decrease of $n_o$ and increase of $n_e$. By contrast, in the dimerized and off-resonant case (**f**), $n_e$ and $n_o$ are conserved independently, indicating an emergent U(1) × U(1) symmetry. The grey dashed lines represent the initial values of $n_o = 6$, $n_e = 10$, $n = 16$ and $n_e + 2n_o = 22$ in **e**. Error bars represent the standard deviation over five rounds of measurements, with each taking 20,000 shots.

between the dynamics of $\{|\Psi_0\rangle\}$ and the echo circuit $U_{echo}(t) = (U^\dagger)^t U^t$ (ref. 44).

To expose the origin of faster decoherence for edge modes at finite temperatures, we introduce excitations into the bulk in a controlled way by initializing the system in the manifold $\{|\Psi_e\rangle\}$ with $n = 16$ excitations. Notably, we observe that $|\Psi_e\rangle$ with excitations near each end can show an even faster decay of the edge modes than the product state (Fig. 2b). To illustrate the effect of excitation positions, we further probe time-dependent expectation values of bulk stabilizers $\{K_i = \sigma_{i-1}^z \sigma_i^x \sigma_{i+1}^z\}_{i=2}^{N-1}$, and edge operators $\{K_1, K_N\} = \{\widetilde{X}_L, \widetilde{X}_R\}$. We define the normalized expectation value as $\overline{\langle K_i \rangle} = \langle \Psi_e | K_i(t) | \Psi_e \rangle / \langle \Psi_0 | K_i(t) | \Psi_0 \rangle$ to underscore the decay caused by excitations. In Fig. 2c, we show the measured $\overline{\langle K_i \rangle}$ dynamics near the left edge with two different initial excitation positions and observe that the edge mode is maintained until excitations propagate to the edge, demonstrating that its rapid decay is due to the edge–bulk interactions.

In the dimerized regime ($J_e \neq J_o$), the edge modes show distinct behaviours (Fig. 2d). Starting with $|\Psi_e\rangle$, we measure the time dependence of edge operators for $J_o/J_e$ ranging from 0.8 to 3.2. It is evident from Fig. 2d that the lifetime of the edge modes is prolonged as $J_o/J_e$ deviates from 1. Theoretically, the edge operators in the dimerized regime can be described as prethermal strong zero modes (Supplementary Information section 1D), which induce almost exact fourfold degeneracy

throughout the entire spectrum, leading to enhanced resilience against thermal excitations[14,35,45]. Taking the left edge as an example, this zero mode, to first order in $h_x$ and $V_{xx}$, is given by[14]

$$\Psi_L^z = \widetilde{Z}_L + \frac{h_x}{J_e}\sigma_1^x \sigma_2^x \sigma_3^z - \frac{V_{xx}}{J_o^2 - J_e^2}(J_e\sigma_1^x\sigma_3^z + J_o\sigma_1^y\sigma_2^y\sigma_3^x\sigma_4^z). \tag{2}$$

Note the divergence of the third term when $J_o/J_e = 1$, where the edge mode has a larger overlap with bulk terms, leading to a shortened lifetime for $\widetilde{Z}_L$. By measuring the late-time average of each operator in $\Psi_L^z$, we experimentally reconstruct[39] the corresponding coefficients and quantitatively verify the theoretical prediction. As shown in Fig. 2e, the measured bulk contribution gradually increases as $J_o/J_e$ decreases from 3.17 to 1.5, signifying that $\Psi_L^z$ extends into the bulk. Moreover, as $J_o/J_e$ decreases further to 0.6, the coefficients of $\sigma_1^x\sigma_3^z$ and $\sigma_1^y\sigma_2^y\sigma_3^x\sigma_4^z$ change sign, as predicted by the analytical expression in equation (2) when crossing the resonance at $J_o/J_e = 1$. Similar divergences arise for $\Psi_L^x$, but with two resonant points ($J_o/J_e = 1, 2$), corresponding to two lifetime dips (Fig. 2d) and two sign changes (Extended Data Fig. 4). This non-monotonicity in the edge mode lifetime and the divergence of coefficients illustrate the intricacy of edge–bulk interactions, providing a distinction between the dimerization mechanism and the suppression of interaction strength.

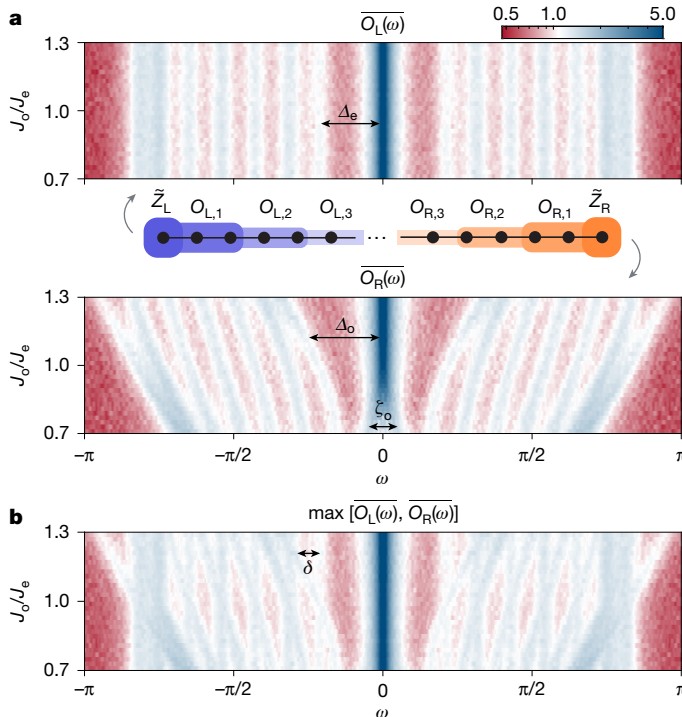

**a**

$\overline{O_{\rm L}(\omega)}$

0.5  1.0  5.0

$\Delta_{\rm e}$

$\tilde{Z}_{\rm L}$   $O_{\rm L,1}$   $O_{\rm L,2}$   $O_{\rm L,3}$   $O_{\rm R,3}$   $O_{\rm R,2}$   $O_{\rm R,1}$   $\tilde{Z}_{\rm R}$

$\overline{O_{\rm R}(\omega)}$

$\Delta_{\rm o}$

$\zeta_{\rm o}$

**b**

$\max[\overline{O_{\rm L}(\omega)}, \overline{O_{\rm R}(\omega)}]$

$\delta$

**Fig. 4 | Spectroscopy of energy spectrum. a**, Averaged Fourier transforms of $\overline{Z}_{\rm L}$ and bulk terms $O_{\rm L,1}$, $O_{\rm L,2}$, $O_{\rm L,3}$ dynamics as a function of $\omega$ and $J_{\rm o}/J_{\rm e}$, revealing the spectrum of the upper Kitaev chain in Fig. 1b (top). Similar Fourier results on the right, revealing the spectrum of the lower Kitaev chain (bottom). The gap $\zeta_{\rm o}$ indicates the hybridization between edge modes. The gaps $\Delta_{\rm o}$ and $\Delta_{\rm e}$ separate the edge modes from the bulk excitation mode. **b**, The complete spectrum obtained from combining $\overline{O_{\rm L}(\omega)}$ and $\overline{O_{\rm R}(\omega)}$, where $\delta$ represents the gap between the bulk modes on different Kitaev chains. The results are obtained from a chain with $N = 16$ qubits.

## Excitation dynamics and emergent symmetry

To understand the dimerization mechanism for enhancing the lifetime of edge modes at finite temperatures, we examine site-resolved excitation dynamics and bulk–edge interactions for the whole chain. We plot the measured dynamics of $\langle \overline{K_i} \rangle$ for $J_{\rm o}/J_{\rm e} = 1.0$ (homogeneous), 2.0 (dimerized but resonant) and 3.17 (dimerized and off-resonant) in Fig. 3 (see also Extended Data Fig. 5 for the raw data). The excitation dynamics are distinct in the three cases. First, in the homogeneous case, excitations deep in the bulk propagate diffusely across even and odd sites (Fig. 3a). By contrast, in the two dimerized cases, excitations initially located at even (or odd) sites are constrained to move along sites of the same parity (Fig. 3b,c). Neighbouring excitation pairs with different parities propagate freely without interacting with each other, whereas pairs with the same parity collide. Second, excitations near the boundaries in the homogeneous case are absorbed by the edge states, whereas for the dimerized and off-resonant case (Fig. 3c), they are reflected at the boundaries without affecting the edge states (see also Extended Data Fig. 6). Third, for the dimerized but resonant case (Fig. 3b), despite similar dynamics observed near the right boundary as in the off-resonant case, the excitations interact strongly with the left edge because of the resonance (Supplementary Information section 1D).

The distinct behaviours of the three cases above can be better understood in the Majorana fermion picture (Fig. 1b and Methods). Through Jordan–Wigner transformation, the cluster Hamiltonian $H_0$ is transformed into two Kitaev chains composed of Majorana fermions on even and odd sites, respectively. The stabilizers centred at even sites are mapped to inter-site coupling terms with strength $J_{\rm e}$ in the upper chain, and odd sites are mapped to inter-site coupling terms with strength $J_{\rm o}$ in the lower chain. The edge mode is mapped to two

Majorana fermions at the end of each Kitaev chain, and the single- and two-body terms in $H_1$ are mapped to onsite and inter-chain coupling terms. With $J_{\rm o} = J_{\rm e}$, the two Kitaev chains share the same coupling strength and can exchange excitations resonantly through $V_{xx}$ terms both in the bulk and at the boundaries, in which the latter couple to the edge Majorana fermions and lead to the decay of the edge modes. In the small-perturbation regime ($h_x, V_{xx} \ll J_{\rm o}, J_{\rm e}$), the system exhibits long-lived prethermal behaviour with an approximate U(1) symmetry of the total excitation number $n$ in the bulk. However, despite the conserved $n$, the number of bulk excitations on even sites $n_{\rm e}$ and odd sites $n_{\rm o}$ rapidly equilibrate in the homogeneous case (Fig. 3d), showing the effect of the resonant inter-chain interactions. Dimerizing the coupling strengths makes the excitation exchange in the bulk off-resonant, but resonances can still arise at the boundaries for certain values of $J_{\rm o}/J_{\rm e}$. For example, when $J_{\rm o}/J_{\rm e} = 2.0$, the exchange of one excitation in the lower chain and two excitations in the upper chain through the $\sigma_2^x \sigma_3^x$ term in $H_1$ becomes resonant. This results in the observed rapid decay of $\widetilde{X}_{\rm L}$, and all $n_{\rm e}$, $n_{\rm o}$ and $n$ are no longer conserved (Fig. 3e). This resonance can be eliminated by choosing $J_{\rm o}$ and $J_{\rm e}$ to be incommensurable. Consequently, the excitation exchanges become off-resonant both in the bulk and at the boundaries, leading to two Kitaev chains effectively decoupled and exhibiting two separate approximate U(1) conservation laws for $n_{\rm e}$ and $n_{\rm o}$ (Fig. 3f). This U(1) × U(1) symmetry emerges in the prethermal regime in which $J_{\rm e}, J_{\rm o} \gg h_x, V_{xx}$ and $J_{\rm o}/J_{\rm e}$ is off-resonant, which is also confirmed in our numerical simulations with matrix product states (Supplementary Information section 1H). The lifetime of this regime scales exponentially with the ratio of the energy scale of $H_0$ to that of the perturbation $H_1$ (Supplementary Information section 1E). Up to this timescale, the U(1) × U(1) symmetry, combined with the inherent $\mathbb{Z}_2 \times \mathbb{Z}_2$ symmetry of the system, gives rise to robust edge modes persisting up to infinite temperature.

## Energy spectrum

Recent theoretical progress suggests that prethermalization is a generic phenomenon in gapped local many-body systems, in which quantum dynamics is restricted to each symmetry sector protected by the energy gaps[46]. This prediction also applies to our experiments, as the emergent U(1) × U(1) symmetry and the robust edge modes are manifestations of energy gaps in the spectrum. Using the energy spectroscopy technique[39,47], we measure the spectrum of smaller SPT chains in the integrable limit ($V_{xx} = 0$) on another processor[48] in parallel, which has a similar design but better coherence performance. To enhance the experimental visibility of the energy spectrum, we measure the time-domain dynamics of a set of operators $O_{\rm L,i} = \left( \prod_{k=1}^{2i} \sigma_k^x \right) \sigma_{2i+1}^z$, $O_{\rm R,i} = \left( \prod_{k=1}^{2i} \sigma_{2N+1-k}^x \right) \sigma_{2(N-i)}^z$ and average their spectra after Fourier transformation, which enables a faithful detection of energy gaps in our experiments for system sizes up to $N = 16$ qubits (Supplementary Information sections 1G and 2D).

For the 16-qubit chain, we measure the dynamics of $O_{\rm L/R,i}(t)$ up to $i = 3$, where $O_{\rm L/R,3}$ are 7-body operators (Supplementary Fig. 5). The averaged frequency-domain signals are shown in Fig. 4a. The results provide substantial information to understand the origin of the emergent symmetries. First, as the two Kitaev chains are decoupled at $V_{xx} = 0$, $\overline{O_{\rm L}(\omega)}$ gives access to the spectrum for the upper chain and $\overline{O_{\rm R}(\omega)}$ gives access to the spectrum for the lower chain. The peaks correspond to Bogoliubov fermionic modes in each chain, in which peaks near $\omega = 0$ are attributed to the edge modes, and the remaining peaks characterize the bulk excitation modes. In our finite-sized system, the edge modes are hybridized by gaps $\zeta$ induced by the $h_x \sigma_i^x$ terms in $H_1$. As we increase $J_{\rm o}$, thereby decreasing the correlation length in the lower Kitaev chain (Fig. 1b), one such gap $\zeta_{\rm o}$ in $\overline{O_{\rm R}(\omega)}$ closes. Furthermore, in Extended Data Fig. 7, we observe that $\zeta_{\rm e}$, $\zeta_{\rm o}$ gradually close as the system size increases. Second, we observe gaps $\Delta_{\rm o} \propto J_{\rm o}$ ($\Delta_{\rm e} \propto J_{\rm e}$) separating the edge mode from the bulk excitation modes, impeding transitions between edge and bulk caused by onsite interactions. When the two chains are

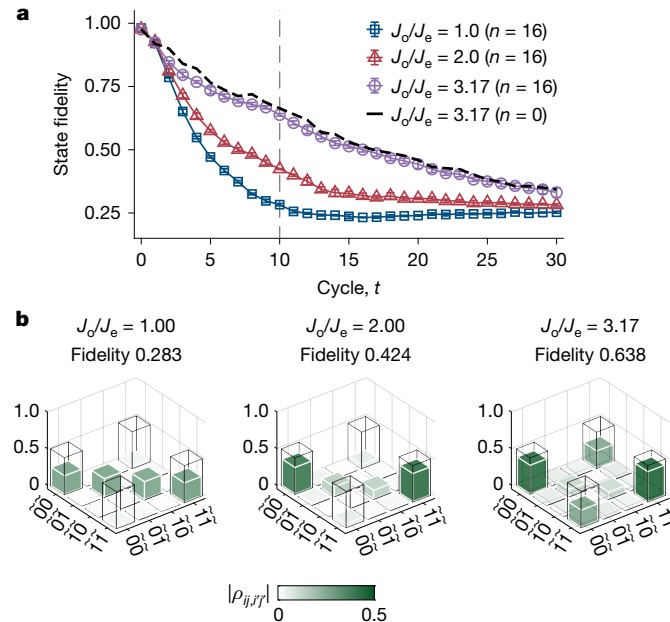

**a**

**b**

$J_o/J_e = 1.00$ Fidelity 0.283 $\quad J_o/J_e = 2.00$ Fidelity 0.424 $\quad J_o/J_e = 3.17$ Fidelity 0.638

$|\rho_{ij,i'j'}|$ 0 0.5

**Fig. 5 | Fidelity dynamics of the logical Bell state at finite temperature.** **a**, Measured fidelity dynamics of logical Bell state in the homogeneous ($J_o = J_e = \pi/5$), the dimerized but resonant ($J_o = 2J_e = 2\pi/5$), and the dimerized and off-resonant ($J_o = 3.17J_e = 3.17\pi/5$) cases. The data shown in the solid lines are obtained with initial states being within $\{|\Psi_e\rangle\}$ and the dashed lines are obtained with initial states being within $\{|\Psi_o\rangle\}$. The initial state preparation circuit is shown in Extended Data Fig. 8a. Error bars represent the standard deviation over five rounds of measurements, with each round taking 10,000 shots for each operator. **b**, Measured density matrices (green bars) of logical Bell state after a time evolution of $t = 10$ in the three different cases with initial states being within $\{|\Psi_e\rangle\}$. The ideal Bell state density matrix is shown with the hollow frame.

decoupled, $\Delta_e$, $\Delta_o$ give rise to an approximate U(1) symmetry in each chain. However, these U(1) symmetries can be destroyed when inter-chain interactions are present. In the full spectrum of the entire system (Fig. 4b) obtained from combining $\overline{O_L(\omega)}$ and $\overline{O_R(\omega)}$, we observe that the bulk energy spectra of the two Kitaev chains become exactly equal when $J_o/J_e = 1$. This explains the strongly resonant excitation exchange process observed in the homogeneous case. When the system is dimerized ($J_o \neq J_e$), an extra gap $\delta \propto |J_e - J_o|$ appears, signifying the energy required to exchange one pair of excitations between the chains. This gap bolsters the emergent U(1) × U(1) symmetry and suppresses the excitation exchange process at boundaries, resulting in robust long-lived edge modes up to infinite temperature. Notably, we find that these gaps, $\Delta_e$, $\Delta_o$ and $\delta$, persist as the system size increases (Extended Data Fig. 7).

## Protection of logical Bell state

The long-lived topological edge modes observed in experiments offer a potential application to store quantum information at finite temperatures. Compared with physical qubits, our approach protects the edge modes from local, symmetry-preserving noises (Supplementary Information section 2F). These edge modes also contrast with the Ising chains, in which a classical bit might be preserved by edge spin polarization[35,39]. To this end, we prepare a logical Bell state encoded by these edge states and show its robustness to thermal excitations. Owing to the geometrically adjacent two edges on the processor (Fig. 1a, blue circles), we can initialize the system with edge modes being a logical Bell state $|\tilde{0}\rangle_L|\tilde{0}\rangle_R + i |\tilde{1}\rangle_L|\tilde{1}\rangle_R$ by local two-qubit gates (see Extended Data Fig. 8 for the details of the preparation circuit and logical Bell state fidelity).

The solid lines in Fig. 5a show the measured fidelity dynamics of the logical Bell state for initial states being within $\{|\Psi_e\rangle\}$ with $J_o/J_e = 1.0$, 2.0 and 3.17, respectively. As expected, the fidelity in the homogeneous scenario decays the most rapidly to the lower bound of 0.25, followed by the dimerized but resonant system. The lifetime of the logical Bell state in the dimerized and off-resonant system is largely prolonged, almost reaching that of the zero-temperature case (dashed line). Furthermore, we carry out state tomography (Supplementary Information section 2E) on the logical space of each system after a time evolution of $t = 10$. As shown in Fig. 5b, the logical Bell state in the homogeneous case is completely decohered, corresponding to an identity matrix of a maximally mixed state. In the dimerized but resonant case, it also exhibits rapid decoherence with vanishing off-diagonal terms. By contrast, it is largely preserved for the dimerized and off-resonant case, showing notable robustness against thermal excitations at finite temperature.

## Discussion

The robust edge modes observed in our experiments are attributed to emergent symmetries within the prethermal regime, thereby eliminating the necessity for strong disorder. We established that these symmetries arise from distinct gaps in the energy spectrum, a common phenomenon in quantum many-body systems. This dimerization-induced prethermalization mechanism is neither restricted to 1D systems nor SPT phases. Recent works predict the robust storage of quantum information at finite temperatures using boundary modes at interfaces between distinct phases[49], nonlocal operators in toric codes[12] and 2D subsystem codes[50], and local corner modes in higher-order SPT phases[51]. The quantum processor developed in this work would also be exploited in these scenarios. In particular, it would be interesting and important to implement boundary modes at phase interfaces and corner modes in higher dimensions. Our work opens new possibilities for quantum information storage resilient to thermal excitations on noisy intermediate-scale quantum devices. Furthermore, it has been shown that periodically and quasi-periodically driven systems possess additional dynamical symmetries, which can supplement or even replace the intrinsic symmetries of the Hamiltonian[38,52,53]. It could be possible to extend our study to realize new dynamical SPT phases that possess edge modes resilient against both perturbations and thermal excitations, without relying on any intrinsic symmetry or localization.

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

## Methods

### Experimental setup

Our experiments are performed on a 2D flip-chip superconducting quantum processor, which possesses 125 frequency-tunable transmon qubits[54] and 218 tunable couplers[55] between the adjacent qubits (Fig. 1a). In our experiments, we actively use 100 qubits and 100 couplers of them to simulate the many-body dynamics of 1D disorder-free cluster Hamiltonian $H$ in equation (1). Each time step of its evolution unitary $U(\delta t)$ is decomposed into combinations of single-qubit rotations and two-qubit gates. For each qubit, a single-qubit rotation is implemented by applying a microwave pulse or a fast flux pulse, which are combined by a combiner at room temperature and then transmitted to the qubit at low temperature (20 mK) to rotate the qubit state along longitudinal or latitudinal lines of the Bloch sphere. Two-qubit interaction between the nearest-neighbour two qubits can be dynamically controlled by applying a fast flux pulse to the corresponding coupler, which also enables the implementation of high-fidelity two-qubit controlled-phase (CPhase) gates[56]. Each qubit is capacitively coupled to a readout resonator for dispersive readout, which is designed at a frequency of around 6.4 GHz. The processor is integrated into a printed circuit board package using the wire bonding technique. This package is further protected by magnetic shields before being mounted on the mixing chamber plate of a dilution refrigerator. See Supplementary Fig. 7 for the wiring information of the dilution refrigerator and room-temperature control electronics.

### Initial state preparation

In our experiments, the system is initialized to the manifold $\{|\Psi_0\rangle\}$ with no excitation, the manifold $\{|\Psi_e\rangle\}$ with 16 excitations, or the product states $|\bullet 00...0\bullet\rangle$, each with a predetermined bulk state and varying edge modes. To measure the temporal dependence of the logical operators $\widetilde{Z}$ and $\widetilde{X}$, we prepare the edge modes into their eigenstates, which are denoted as $|\tilde{0}\rangle, |\tilde{1}\rangle$ for $\widetilde{Z}$, and $|\tilde{+}\rangle, |\approx\rangle$ for $\widetilde{X}$. For the zero-temperature case, these states are defined as

$$|\tilde{0}\rangle_L|\tilde{0}\rangle_R = \prod_{i=1}^{99} CZ_{i,i+1}\left[|0\rangle_1\left(\bigotimes_{i=2}^{99}|+\rangle_i\right)|0\rangle_{100}\right], \quad (3)$$

$$|\tilde{+}\rangle_L|\tilde{+}\rangle_R = \prod_{i=1}^{99} CZ_{i,i+1}\left(\bigotimes_{i=1}^{100}|+\rangle_i\right), \quad (4)$$

and the circuits for preparing these states are shown in Extended Data Fig. 2a,b. For $\{|\Psi_e\rangle\}$, excitations are induced into the bulk by applying $X_i(\pi)$ ($Z_i(\pi)$ in Extended Data Fig. 6) gates on the qubit $i$. For the product-state case, we prepare the $|000...00\rangle$ state for measuring $\{\widetilde{Z}_L, \widetilde{Z}_R\}$ and the $|+00...0+\rangle$ state for measuring $\{\widetilde{X}_L, \widetilde{X}_R\}$.

The preparation for the logical Bell state $|\tilde{0}\rangle_L|\tilde{0}\rangle_R + i |\tilde{1}\rangle_L|\tilde{1}\rangle_R$ is more involved. This is done by first applying a logical $\widetilde{X}(-\pi/2)$ rotation on $|\tilde{0}\rangle_L|\tilde{0}\rangle_R$, and then a combination of two-qubit gates and single-qubit gates on two edge modes to effectively implement the logical controlled-NOT gate. The total circuit for preparing the logical Bell state is shown in Extended Data Fig. 8a.

### Characterization of Trotter errors

Quantum simulation of continuous-time many-body dynamics with a discretized evolution circuit $U$ is prone to an accumulation of Trotter errors, which tends to heat the system to infinite temperature. However, with a small Trotter step, the heating is suppressed by the Floquet prethermalization, leading to an exponentially long heating time $t_*$. For $t < t_*$, the stroboscopic dynamics of system are governed by an effective Hamiltonian $H_F$, defined by $\exp(-iH_FT) \equiv U$. Although $H_F$, in general, is difficult to analyse, it can be constructed order by order by the Floquet–Magnus expansion[57,58], in which the lower-order terms are

sufficient to describe the short-term evolution on current noisy intermediate-scale quantum devices. The zeroth-order term gives $H_0 + H_1$, and the first-order terms present many other many-body terms (Supplementary Information section 1B). Hence, these additional terms can be considered as extra interactions and make the edge–bulk interaction in our model more general.

### Transformation to Majorana fermions

The spin Hamiltonian $H = H_0 + H_1$ can be transformed into two Kitaev chains of Majorana fermions. This is done by first applying the Jordan–Wigner transformation, which maps Pauli spin operators into fermionic creation and annihilation operators, and then transforming the latter into Majorana operators $\alpha_i, \beta_i$ (Supplementary Information section 1F). The total transformation reads

$$\sigma_i^x = -i\alpha_i\beta_i, \quad \sigma_i^z = -\left[\prod_{j=1}^{i-1}(-i\alpha_j\beta_j)\right]\alpha_i. \quad (5)$$

Besides $\sigma_i^x$, the three-body stabilizers and two-body interactions in $H$ are mapped into the following forms:

$$\sigma_{i-1}^z\sigma_i^x\sigma_{i+1}^z = -i\beta_{i-1}\alpha_{i+1}, \sigma_i^x\sigma_{i+1}^x = -\alpha_i\beta_i\alpha_{i+1}\beta_{i+1}. \quad (6)$$

Notably, the three-body stabilizers at even sites are mapped into coupling terms involving Majorana operators only at odd sites and those at odd sites are mapped into coupling terms involving Majorana operators only at even sites, giving rise to two Kitaev chains. Moreover, the logical operators for edge modes become

$$\widetilde{Z}_L = -\alpha_1, \widetilde{X}_L = -\alpha_2, \widetilde{Z}_R = -iG\beta_N, \widetilde{X}_R = -iG\beta_{N-1}, \quad (7)$$

where $G = \prod_{j=1}^N(-i\alpha_j\beta_j) = \prod_{j=1}^N \sigma_j^x$, is the generator for the total $\mathbb{Z}_2$ symmetry. As $H$ preserves the $\mathbb{Z}_2 \times \mathbb{Z}_2$ symmetry generated by $\prod_{i=1}^{\frac{N}{2}}\sigma_{2i}^x$ and $\prod_{i=1}^{\frac{N}{2}}\sigma_{2i-1}^x$, $G$ is also preserved during the evolution. Therefore, the logical operators $\widetilde{Z}_L, \widetilde{X}_L, \widetilde{Z}_R$, and $\widetilde{X}_R$ are determined by Majorana edge modes $\alpha_1, \alpha_2, \beta_{N-1}$, and $\beta_N$, respectively.

### Data availability

The data that support the findings in this study are available on Code Ocean[59].

### Code availability

The code used in this study is available on Code Ocean[59].

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

**Acknowledgements** We thank A. Gorshkov, F. L. Liu and Z. X. Gong for discussions. The device was fabricated at the Micro-Nano Fabrication Center of Zhejiang University. The electronics for controlling the superconducting quantum processors were developed by Hangzhou Logical Qubit Technology. We acknowledge the support from the National Natural Science Foundation of China (grant nos. T2225008, 92365301, 12274368, 12274367, 12174342, 12322414, 12404570, 12404574, U20A2076, 12075128 and 123B2072), the Innovation Program for Quantum Science and Technology (grant nos. 2021ZD0300200 and 2021ZD0302203), the Zhejiang Provincial Natural Science Foundation of China (grant nos. LR24A040002 and LDQ23A040001) and the National Key Research and Development Program of China (grant no. 2023YFB4502600). H. Wang is also supported by the New Cornerstone Science Foundation through the XPLORER PRIZE. T.I. acknowledges support from the National Science Foundation (grant no. DMR-2143635). F.M. acknowledges support from the NSF through a grant for ITAMP (award no. 2116679) at

Harvard University. J.K. acknowledges support from the Army Research Office (grant no. W911NF-24-1-0079) and from EPSRC (grant no. EP/V062654/1). N.Y.Y. acknowledges support from the US Department of Energy by the QuantISED 2.0 program and from a Simons Investigator award. S.J., W.L., Z.L., Z.-Z.S. and D.-L.D. acknowledge in addition support from the Tsinghua University Dushi Program and the Shanghai Qi Zhi Institute Innovation Program SQZ202318.

**Author contributions** F.J., X.Z. and Z.B. carried out the experiments and analysed the experimental data under the supervision of Q.G. and H. Wang; S.J., J.K., N.Y.Y., T.I., F.M., W.L., Z.L., Z.-Z.S., D.Y. and D.-L.D. conducted the theoretical analysis; H.L. and J.C. fabricated the device supervised by H. Wang; D.-L.D., Q.G., S.J., F.J., X.Z., H. Wang, J.K., N.Y.Y., T.I. and F.M. co-wrote the paper; H. Wang, Q.G., Z.W., C.S, J. Zhang, F.J., X.Z., Z.B., F.S., K.W., Z.Z., S.X., Z.S., J.C., Z.T., Y. Wu, C.Z., Y.G., N.W., Y.Z., A.Z., T.L., J. Zhong, Z.C., Y. Han, Y. He, Han Wang, J.-N.Y., Y. Wang, J.S., G.L., J.D., H.D. and P.Z. contributed to the experimental setup. All authors contributed to the discussion of the results.

**Competing interests** The authors declare no competing interests.

**Additional information**
**Correspondence and requests for materials** should be addressed to Qiujiang Guo, H. Wang or Dong-Ling Deng.

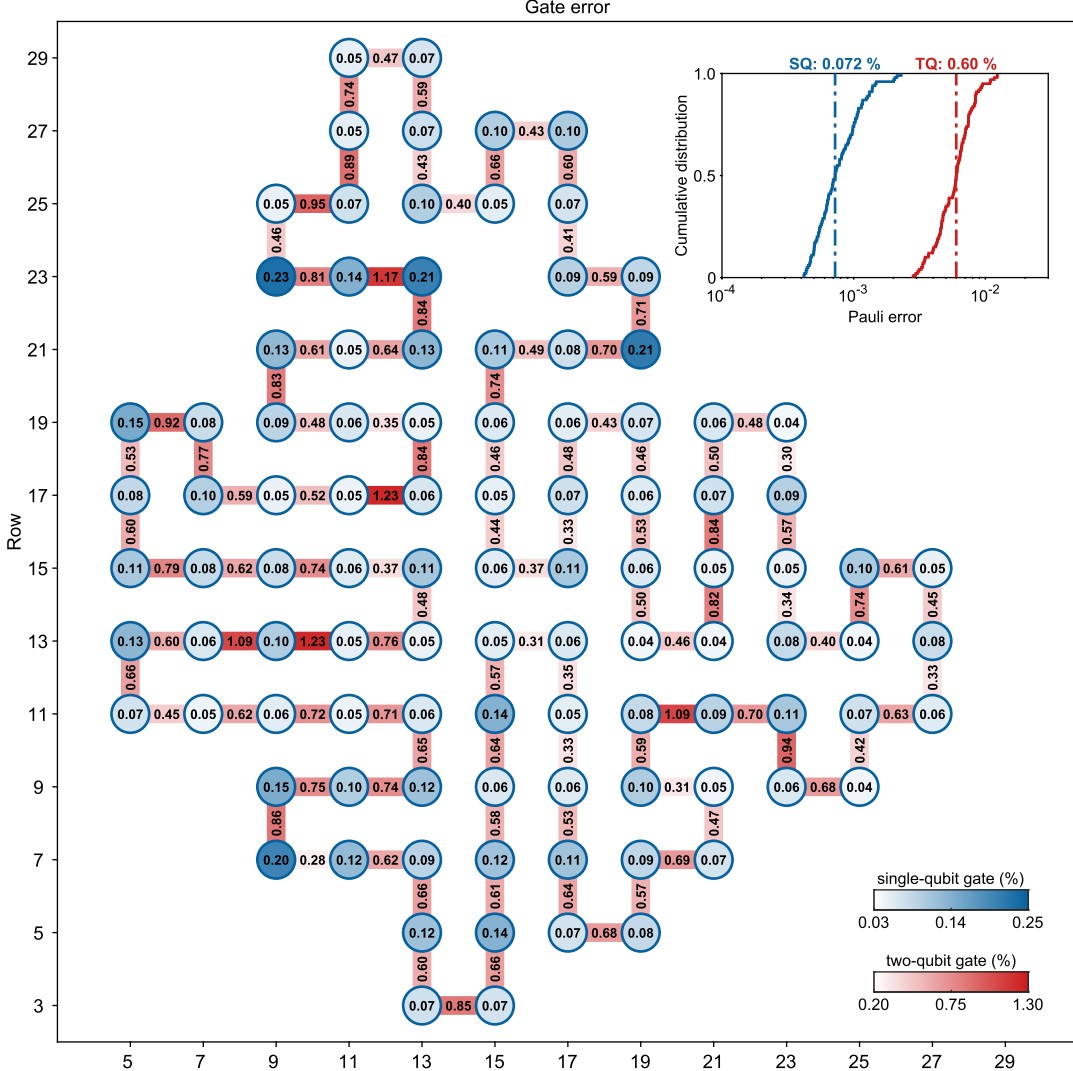

**Extended Data Fig. 1 | Pauli errors of single-qubit and two-qubit gates.**
Gate errors are benchmarked with simultaneous cross-entropy benchmarking (XEB). Errors of single-qubit gates (blue circles) are obtained by running single-qubit XEB sequences for all 100 qubits simultaneously, while errors of two-qubit gates [red bars, including CZ and CPhase (−0.4)] are averaged over the two-qubit layers used in our experiments. For each two-qubit layer, we run two-qubit XEB sequences simultaneously for all the two-qubit gates in this layer. The maximum number of parallel two-qubit gates in our experiments is 50. The inset shows the cumulative distribution of gate errors, with the dashed lines indicating the median values.

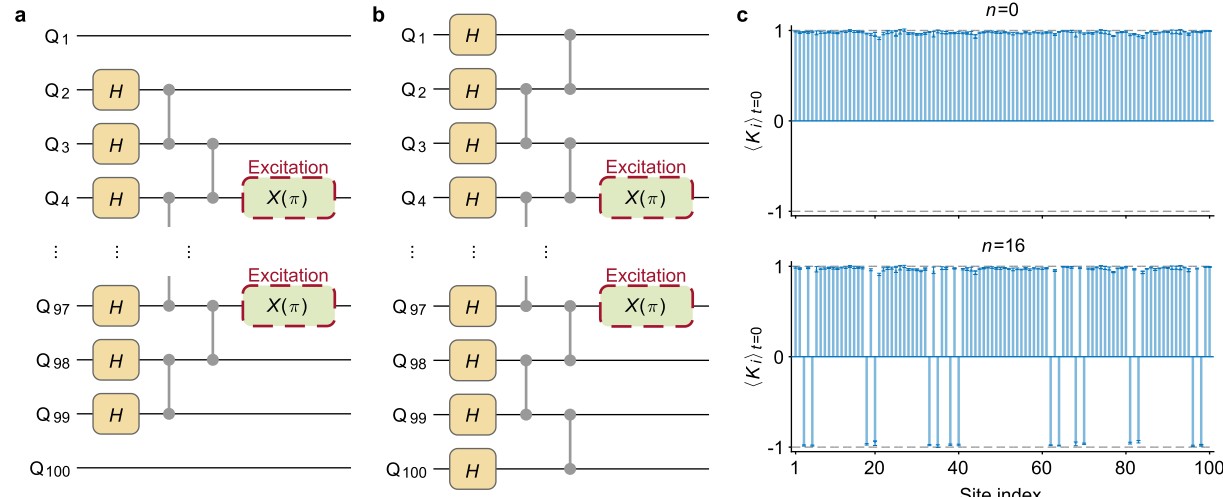

**Extended Data Fig. 2 | Initial state preparation. a**, Quantum circuit for preparing the initial cluster state for measuring $\langle \overline{Z}(t) \rangle$ in Fig. 2b,d and Extended Data Fig. 3a,c. The first three layers, including one layer of Hadamard gates and two layers of CZ gates, prepare the zero-temperature state in $\{|\Psi_0\rangle\}$ with all stabilizers and $\overline{Z}_L, \overline{Z}_R$ taking values +1. The last layer applies single-qubit $\pi$ rotations around the $x$-axis of the Bloch sphere [$X(\pi)$ gates] on the bulk qubits, each flipping two stabilizers and hence inducing two bulk excitations. The initial states in the manifold $\{|\Psi_e\rangle\}$ are obtained from applying the $X(\pi)$ gates on qubits $\{Q_4, Q_{19}, Q_{34}, Q_{39}, Q_{63}, Q_{69}, Q_{82}, Q_{97}\}$, which induces 16 excitations in the bulk. In the bottom panel of Fig. 2c and Extended Data Fig. 3b, the $X(\pi)$ gates are

applied to $\{Q_6, Q_{19}, Q_{34}, Q_{39}, Q_{63}, Q_{69}, Q_{82}, Q_{95}\}$ to observe the effect of varying excitation positions on the edge modes. **b**, Quantum circuit for preparing the initial state for measuring $\langle \overline{X}(t) \rangle$ in Fig. 2, Fig. 3 and Extended Data Fig. 3, with $\overline{X}_L, \overline{X}_R$ taking values +1. **c**, Expectation values of the stabilizers $K_i$ ($\sigma_{i-1}^z \sigma_i^x \sigma_{i+1}^z$ in the bulk and $\overline{X}_L, \overline{X}_R$ at the edges) for initial states in $\{|\Psi_0\rangle\}$ (top panel) and in $\{|\Psi_e\rangle\}$ (bottom panel). The data are extracted from Extended Data Fig. 5 at $t = 0$. Error bars represent the standard deviation over fifteen rounds of measurements (five for the homogeneous case, five for the dimerized but resonant case, and five for the dimerized and off-resonant case), with each taking 20,000 shots. Grey dashed lines indicate the values of ±1.

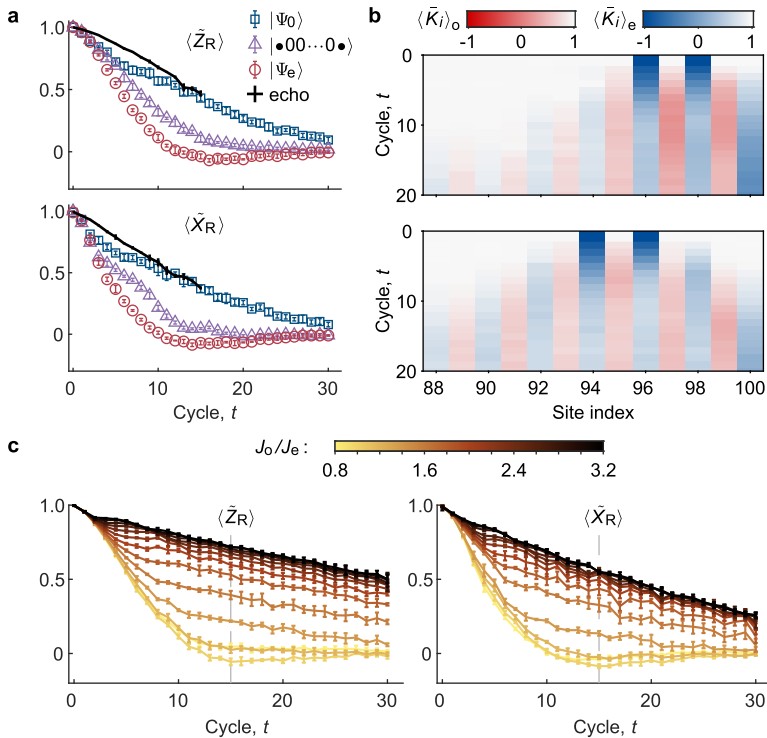

**Extended Data Fig. 3 | Measured time dynamics of the right edge operators.**
**a**, $\langle \tilde{Z}_R(t) \rangle$ and $\langle \tilde{X}_R(t) \rangle$ in the homogeneous case ($J_o = J_e = \pi/5$). **b**, Measured site-resolved dynamics of normalized expectation value $\overline{\langle K_i \rangle}$ near the right edge. The nearest excitations to the right edge are initialized at $\{Q_{96}, Q_{98}\}$ (top panel) and $\{Q_{94}, Q_{96}\}$ (bottom panel). **c**, Measured $\langle \tilde{Z}_R(t) \rangle$ and $\langle \tilde{X}_R(t) \rangle$ with fixed $J_e = \pi/5$ and varying $J_o$. Error bars in **a** and **c** represent the standard deviation over five rounds of measurements, with each taking 10,000 shots.

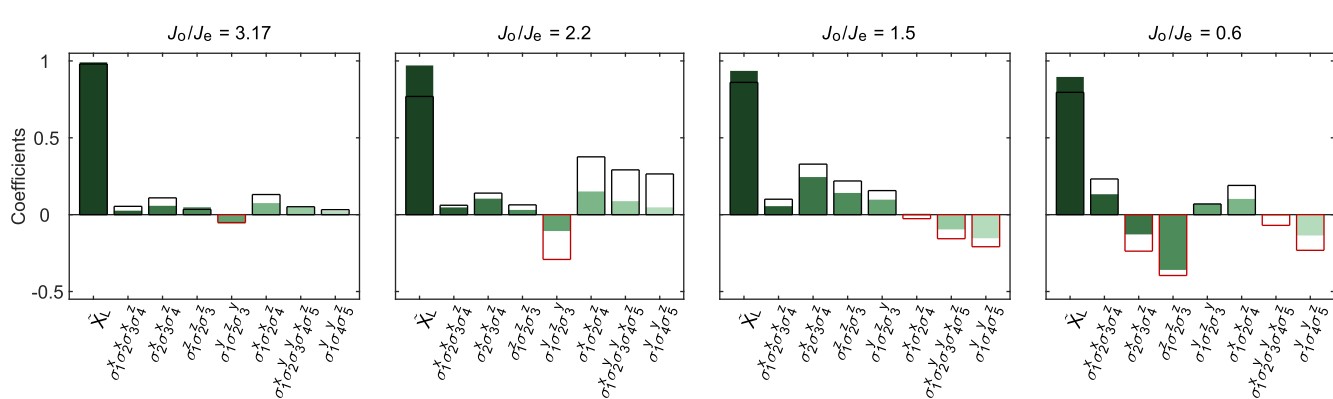

$$\Psi_{\mathrm{L}}^{x} = \tilde{X}_{\mathrm{L}} + \frac{h_x}{J_o}\sigma_1^x\sigma_2^x\sigma_3^x\sigma_4^z + \frac{V_{xx}}{J_o^2 - J_e^2}\left(J_o\sigma_2^x\sigma_3^x\sigma_4^z + J_e\sigma_1^z\sigma_2^z\sigma_3^z\right) - \frac{V_{xx}J_e}{J_o^2 - 4J_e^2}\left[\sigma_1^y\sigma_2^z\sigma_3^y + \left(\frac{2J_e}{J_o} - \frac{J_o}{J_e}\right)\sigma_1^x\sigma_2^x\sigma_4^z - \sigma_1^x\sigma_2^y\sigma_3^y\sigma_4^x\sigma_5^z - \frac{2J_e}{J_o}\sigma_1^y\sigma_4^y\sigma_5^z\right]$$

**Extended Data Fig. 4 | Prethermal strong zero mode $\Psi_{\mathrm{L}}^{x}$ and its spatial profile.** $\Psi_{\mathrm{L}}^{x}$ diverges at both $J_o/J_e = 1$ and 2, where we observe the changes of signs for operators when crossing these two points. The solid boxes denote theoretical predictions, where black and red frames mark positive and negative values, respectively. The coefficients are obtained by averaging the late-time dynamics over cycles from $t = 25$ to 40, with the sum of their squares normalized to unity.

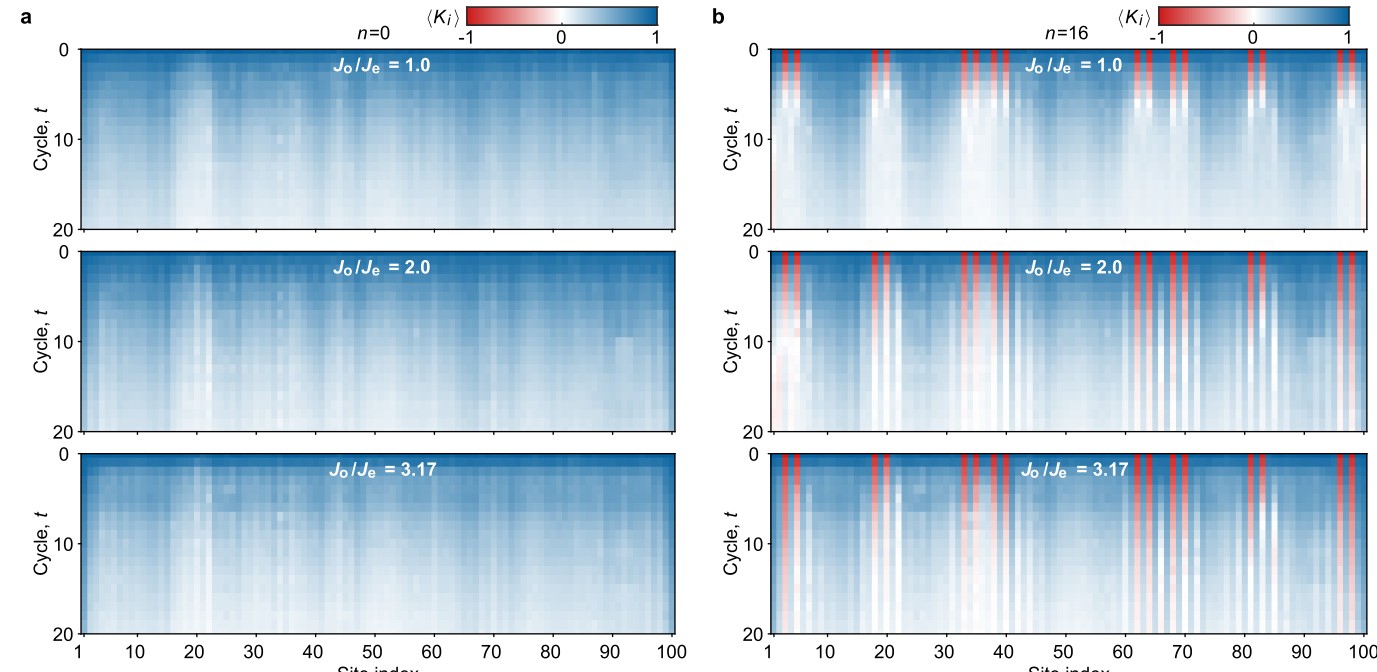

**Extended Data Fig. 5 | Raw data of excitation dynamics. a**, Measured site-resolved dynamics $\langle\Psi_0|K_i(t)|\Psi_0\rangle$ with the initial state being within $\{|\Psi_0\rangle\}$ with no excitation ($n = 0$). The top, middle, and bottom panels show the data obtained from the system in the homogeneous, dimerized but resonant, and dimerized and off-resonant regimes, respectively. **b**, Measured site-resolved dynamics $\langle\Psi_e|K_i(t)|\Psi_e\rangle$ with the initial state being within $\{|\Psi_e\rangle\}$ with $n = 16$ excitations. The excitation dynamics in Fig. 3 are normalized by $\overline{\langle K_i\rangle} = \langle\Psi_e|K_i(t)|\Psi_e\rangle/\langle\Psi_0|K_i(t)|\Psi_0\rangle$ to reveal the effect caused by the bulk excitations.

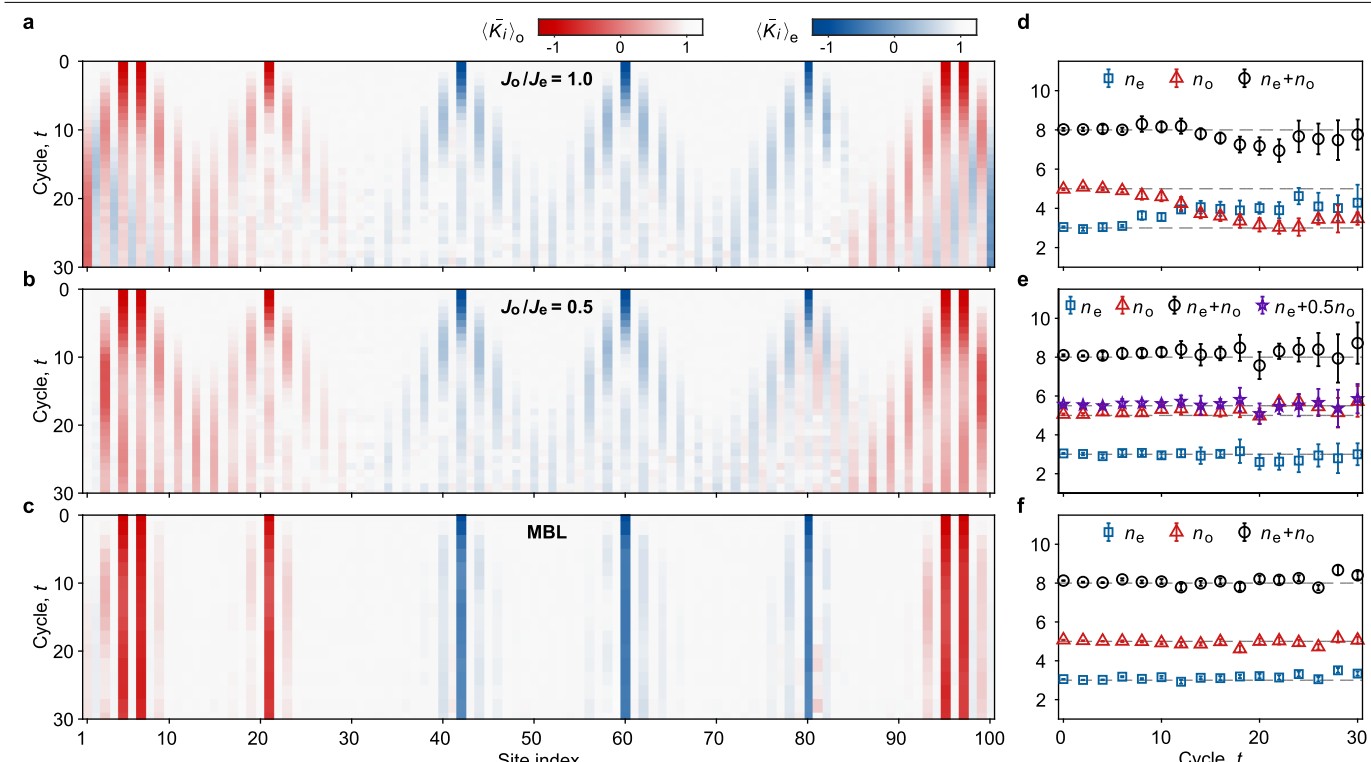

**Extended Data Fig. 6 | Excitation dynamics with only two $V_{xx}$ interaction terms $\sigma_1^x \sigma_2^x$ and $\sigma_{99}^x \sigma_{100}^x$ at the two edges. a-c**, Measured site-resolved dynamics of normalized expectation value $\overline{\langle K_i \rangle}$ for bulk stabilizers $\{\sigma_{i-1}^z \sigma_i^x \sigma_{i+1}^z\}$ and edge operators $\{\widetilde{Z}_L, \widetilde{Z}_R\}$ in the homogeneous (top panel, $J_o = J_e = \pi/2$), dimerized (middle panel, $J_o = 0.5 J_e = \pi/4$), and many-body localized regimes (bottom panel). The bulk excitations are induced by applying $Z(\pi)$ gates on sites $\{Q_5, Q_7, Q_{21}, Q_{42}, Q_{60}, Q_{80}, Q_{95}, Q_{97}\}$. For the system in the homogeneous and dimerized regimes, we set $h_x = 0.23$, $V_{xx} = 0.2$. For the system in the many-body localized regimes, we fix

$V_{xx} = 0.2$ and randomly choose $J_o, J_e, h_x$ from $[\pi/6, 5\pi/6]$, $[\pi/6, 5\pi/6]$, and $[0.18, 0.28]$, respectively for 10 random instances. **d-f**, Measured time dynamics of the total excitation number $n$, excitation number at even ($n_e$) and odd ($n_o$) sites, which are calculated from **a-c**. Error bars in **d** and **e** represent the standard deviation over five rounds of measurements, with each taking 20,000 shots. Error bars in **f** are the standard error of the statistical mean for the 10 instances, with each taking 100,000 shots. Grey dashed lines represent the initial values of $n_o = 5$, $n_e = 3$, $n = 8$ and $n_e + 0.5 n_o = 5.5$ in **e**.

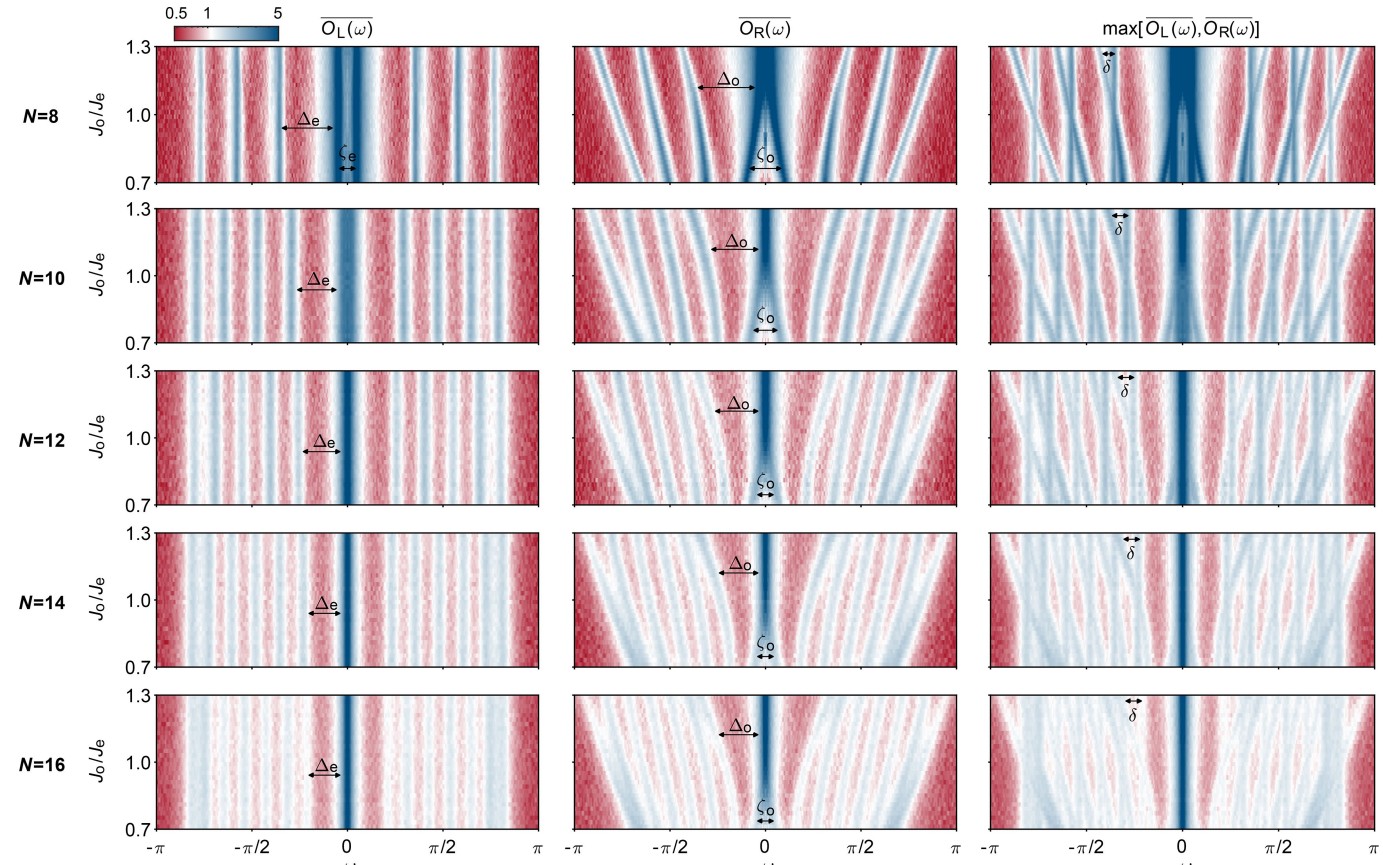

**Extended Data Fig. 7 | Spectroscopy of energy spectra for different system sizes.** Experimental results for system sizes from $N = 8$ to $N = 16$ are displayed from top to bottom. The first two columns show the averaged spectra of the upper and lower Kitaev chains as functions of $\omega$ and $J_o/J_e$. The third column shows their combination as the spectra of the entire system. The presented spectra are obtained by averaging the Fourier spectra of different operators $O_i$, where the averaged sets are chosen to be $i \in \{0\}$ (which are $\overline{Z}_L, \overline{Z}_R$) for $N = 8$, $i \in \{0, 1, 2\}$ for $N = 10, 12$, and $i \in \{0, 1, 2, 3\}$ for $N = 14, 16$.

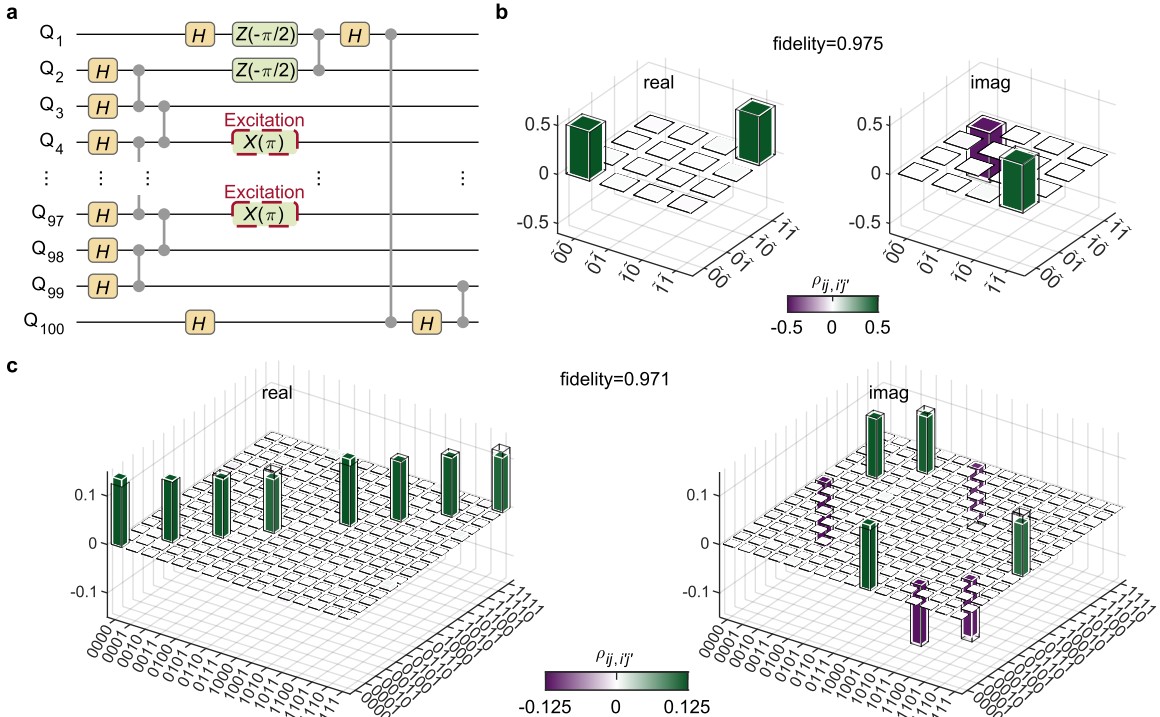

**Extended Data Fig. 8 | Preparation of logical Bell state. a**, Quantum circuit for preparing the logical Bell state $|\tilde{0}\rangle_L|\tilde{0}\rangle_R + i\ |\tilde{1}\rangle_L|\tilde{1}\rangle_R$. The CZ gate applied on two edge qubits $Q_1$ and $Q_{100}$ is local as the two edge qubits are geometrically near to each other on our processor (see Fig. 1 of the main text). **b**, Measured density matrix of the prepared logical Bell state, which is extracted from logical state tomography. Its fidelity is about 0.975. **c**, Measured density matrix in full computational space of $Q_1$, $Q_2$, $Q_{99}$, $Q_{100}$, with fidelity of 0.971 to the ideal density matrix. In **b** and **c**, solid bars are experimental data and hollow frames are the ideal density matrix.