## [Peer Review File · Nature]

Topological prethermal strong zero modes on superconducting processors

Corresponding Author: Professor Dong-Ling Deng

Version 0:

Reviewer comments:

Referee #1

(Remarks to the Author)

In this work the author's report the realization of long-lived finite-temperature edge modes in a disorder-free 125-qubit quantum processor – with 100 of the qubits arranged into a one-dimensional chain.

In order to achieve this, the authors implement a deep quantum circuit which realizes the discrete-time dynamics of a complex Hamiltonian with three-body interactions. The remarkably low gate errors available on the processor allow the quantum dynamics of the system to faithfully emerge even after a significant number of cycles.

The significance and import of the work is very clear: experimentally realizing finite-temperature edge states in a disorder-free system. I am unaware of experimental work realizing such states until now, and achieving this with a digital processor really pushes the boundaries of what is possible with digital quantum simulation.

Due to the above, I do recommend publication of the manuscript. I do, however, have some pertinent comments about the manuscript that I think should be addressed:

Comments

1. It is not clear in the main text how measurements are performed i.e., how many readout shots are taken. What does “five repetitions of measurements” mean in the context of the error bars in the figures? I presume a large number of shots were taken five times and the resulting five shot-averaged values were averaged again and the standard error on the mean was computed to get the error bars? This should be made clearer, especially as the readout error of the device is 88% and this is not reflected in the error bars – which stem solely from a very small statistical sample.
2. I am surprised the authors have not verified / corroborated some of their results with Matrix Product State simulations of the dynamics. These should be straightforward to implement, and it would be interesting to help understand a) the growth of entanglement in the system and b) the faithfulness of the device at realizing the dynamics – at least on short / moderate timescales (depending on how fast entanglement grows and how difficult the simulations are to converge). These simulations would also help to verify the observation of an emergent $U(1) \times U(1)$ that the authors report.
3. In relation to the above, it would seem that the structure of the device means it may be capable of simulating the dynamics of a similar Hamiltonian but in two-dimensions. Can the authors comment further on the feasibility of this with their device and whether they would expect similar physics to emerge? This is briefly alluded to in the conclusion, but it is unclear whether the current device can realize such dynamics or further engineering is required.

(Remarks on code availability)

The data and plotting code used to produce the results is accessible and easy to run due to being well labelled and in the form of `.ipynb` with the corresponding data stored as Excel sheets.

Although, there is no README file for installation / running the code, so I think a short README to describe the structure of

the repository and how users less familiar with the .ipynb format can run the code would be helpful.

Referee #2

(Remarks to the Author)

In the manuscript "Observation of topological prethermal strong zero modes" the authors report on a quantum simulation in a 125 qubit quantum processor, in which they implement quantum dynamics described by a Hamiltonian that realizes symmetry protected topological states in its ground states. Such states have robust boundary modes and the authors probe this boundary mode and show that it can be made long lived depending on the choice of parameters in the dynamics. The manuscript is clearly written and enjoyable to read. It is a nice work that explores interesting topics of currently high interest.

While the work is nice to read, I find it to be rather short on quantitative detail. The theory is used mainly to tell a nice story about the data obtained from the quantum simulator but the detailed analysis of the data and discussion of theoretical consideration that would actually support the claim made in the title, namely that they actually observe topological prethermal strong zero modes, is rather minimal. I miss a detailed and honest discussion of the relevant time scales and what would really be required to experimentally observe what is claimed to observe. In its current form, the manuscript mostly nicely explains the theory, and they show nice data that shows the presence of boundary modes whose coherence time can be controlled by the parameters of the quantum circuit that is applied, but a quantitative connection between the two is missing. This gives the impression that the evidence for the prethermal strong zero modes is rather weak.

For sure the manuscript needs to improve in terms of the details and to make sure that they explain those details to the reader, so as not to mislead the reader. Let me give a few comments in this regard, not necessarily in any specific order.

*) Relevance of the number of qubits N and connection to relevant time scales. What role does N play in theory and experiment? One would want N to be large enough that the overlap of the boundary modes is negligible. This overlap is typically exponential in N and therefore the large number of qubits gives a times $t_{\text{boundary}} \sim \exp(N)$ that is very large such that one does not need to worry about the boundary mode overlap on the time scale of the experiment $t_{\text{exp}} \ll t_{\text{boundary}}$. Does N play any other role?

*) The time scale of the experiment is set by imperfections in the gates and is measure by the echo experiments for example in Fig. 2. We are forced to have $t_{\text{exp}} \sim t_{\text{echo}}$, as beyond that time we don't have coherent quantum dynamics. In the experiment $t_{\text{exp}} \sim 20$ and assuming diffusive dynamics and the fact that $\sqrt{20} \sim 5$, the quantum state spread about 10 qubits on the time scale of the experiment. Therefore when probing the boundary modes on that time scale, most of the $N = 100$ qubits are kind of irrelevant, all the dynamics on that time scale is happening at the boundary. One might guess that doing the same experiment with $N = 20$ or less would probably give the same results. And in fact, in Fig. S3 the authors compare their experiments with numerical simulations in systems with $N = 14$ and get a reasonable agreement. This is kind of clear, for example, in Fig. 3, where one can see that most of the excitations in the bulk of the system have no time to get anywhere near the boundary and are therefore irrelevant to the physics on the experimental time scale.

*) Usually I think of prethermalization to happen in driven systems. Driven systems tend to heat up to infinite temperature and only when there are some approximate conservation laws one can obtain an exponentially large time scale t_{pre} before which the system reaches a different prethermal state and only after much larger time scale does it thermalize to infinite temperature. Most of the discussion in the paper however is in terms of Hamiltonian dynamics which, since it conserves energy, does not have the same infinite temperature phenomena. Now in the experiment the authors don't actually implement the Hamiltonian dynamics but instead Trotterize the dynamics and through this actually effectively get a driven system. So the physic of prethermalization enters through this Trotterization. This is not clearly explained in the manuscript and the discussion of the physics of prethermal states could be more clearly explained and the relevant time scales explained and given explicitly. Some of this can be kind of obtained from the supplemental material, but even there they authors are not careful in clearly separating the discussion of the Hamiltonian dynamics and the driven system dynamics and what facts belong to which case.

*) Once such a discussion is given and the relevant time scales are given, I suspect that $t_{\text{pre}} \gg t_{\text{exp}}$. Which results in the question of in what sense do the authors actually see prethermalization? There is no plateauing of expectation values of boundary modes that one could interpret as the system reaching a prethermal state. Instead one simply sees decay. This needs to be discussed and explained in the main text.

*) Related to the last point, since $t_{\text{exp}} \ll t_{\text{pre}}, t_{\text{boundary}}$, in which sense does the experiment actually demonstrate symmetry protected topological state?

*) How does the data give evidence for the presence of a strong zero mode, instead of simply a boundary mode? I'm missing quantitative details in this regard.

So, in summary, I feel the main text of the manuscript needs to be a bit more quantitative in analysing all relevant time scales and if they want to argue that what they observe is actually strong prethermal zero modes they need to argue how prethermalization arises in this system and give evidence that they have reached the prethermal state. Some motivation for how one can experimentally verify the presence of strong zero modes versus just a boundary state would also help improve the paper. Of course, the theoretical arguments for why one should observe a prethermal strong zero modes in this system is pretty robust and well known. While the paper describes this theory nicely (apart from some details as discussed above)

there is in principle nothing new in terms of the physics here. My main concern with the manuscript is that it's not clear to me that this physics is accessible on the time scale allowed by the experiment.

(Remarks on code availability)

Referee #3

(Remarks to the Author)

This manuscript "Observation of topological prethermal strong zero modes" by Jin et al. presents an experimental study of a stabilizer Hamiltonian which can be mapped to two coupled Majorana chains. The four-fold degeneracy of the ground state gives rise to two coupled edge modes on each end of the chain. The authors prepare bulk excitations by flipping some of the stabilizers away from the edges, propagating the excitations using an interaction term, and measuring edge modes over time. For specific choices of model parameters, the authors report that the edge modes are largely unaffected by the bulk excitations due to a prethermalization mechanism. Lastly, the authors prepared Bell states using the edge modes and show that they also decay slowly under the same choice of model parameters.

The possibility of long-lived edge modes in stabilizer models, and particularly the potential of using them as qubits, is of interest to the quantum simulation and quantum information community. However, since much of the theoretical finding of this work has already been published [e.g. Ref. 14], my evaluation will apply primarily to the experimental demonstration of the edge mode effect. To this end, I find the current manuscript rather lacking in terms of depth. The figures are largely demonstrations of known theoretical results using already established experimental techniques (spectroscopy, echo etc). There is little effort to illustrate any unique insights coming from performing these tasks on an experimental quantum processor. Let me elaborate:

1. The system size of 100 qubits in this study seems impressive at first sight. It is disappointing to then see that the experiment does little to utilize this system size, or even illustrates that the edge modes are truly many-body in nature. For instance, there is no measurement of spatial profile of the edge modes to show that they in fact extend beyond the edge qubits under the authors' choice of model parameters. If the spatial profile does not extend beyond more than a few qubits, then it begs the question of whether having such a large system size is truly meaningful and if the experimental data will look effectively identical with just a dozen qubits.

2. The time dynamics for all of the figures are very short (20 to 30 cycles). Given the local nature of the Hamiltonian, the lightcone of 40 to 60 qubits in the middle of the chain does not even cover the edge qubits and therefore has zero impact whatsoever. This can be easily seen in the experimental data shown in Fig. 3, where the bulk excitations between sites 25 and 75 simply do not have enough time to spread to the edges. This again begs the question whether the large system size used in this experiment is truly meaningful, and whether the experimental findings are effectively reproducible with a very small system size that one can easily study with a laptop.

3. A main practical selling point, which the authors hint at using Fig. 5, is the usage of edge modes as a quantum memory. However, the manuscript does not explore at all why this quantum memory is advantageous to a single uncoupled physical qubit. What does this complicated many-body model buy us in terms of noise protection? Is there a noise direction to which the edge modes are insensitive but physical qubits are? Can the authors demonstrate this advantage in experiment?

4. Relatedly, there is no exploration or discussion of the role of extrinsic decoherence and whether that affects the edge modes differently compared to a physical qubit. For example, how does the Bell state decay rate in Fig. 5 compare with what you could achieve with just those two edge qubits alone without the other 98 qubits? If it's not any better, then why would we adopt such a convoluted scheme?

Without addressing these fundamental questions, I feel that the manuscript will not deliver the impact that a quantum simulation work in Nature should aim at delivering, especially given the preponderance of closely related edge mode papers that have already been published (e.g. Refs 37 and 39).

Some other more specific questions:

1. Within the introduction of the manuscript, the authors wrote "This enables us to successfully implement the dynamics of a prototypical SPT Hamiltonian (Fig. 1b) in different regimes with quantum circuits of depth exceeding 270 and gate counts above 17, 000..." This is highly misleading. The majority of the gates do not contribute at all due to the short depth and limited lightcones of the circuits, and the rest of the gates have little weight on the observables due to the slow spreading of excitations (and therefore noise as well).

2. Fig. 2b: The echo experiments drop lower than the actual edge modes, suggesting there is miscalibration in the experiment or some other coherence errors. I therefore find it difficult to accept this reference experiment as evidence that the observed decay is due to only decoherence.

3. Fig. 4: Why reduce the system size to just 8 qubits in this case? I understand 100-qubit spectroscopy is difficult, but it

seems that the authors should at least attempt to measure the spectra for a few system sizes up to ~ 20 and see that the gap Δ responsible for prethermal protection does not disappear at large system sizes.

4. I'm a little confused how it could be that X_L and Z_L can simultaneously start off at 1 (in e.g. Fig. 2b). Shouldn't they form a vector of length at most unity on the Bloch sphere?

(Remarks on code availability)

Version 1:

Reviewer comments:

Referee #1

(Remarks to the Author)

I am pleased to see the authors have taken my suggestions onboard, most notably including Matrix Product State simulations of the 100 qubit system in the appendix.

It is curious, however, that these are not mentioned in the main text. It appears from the Supplemental that these classical simulations are able to resolve the dynamics of the system at hand (on the timescales considered in the experiment -- 100 qubits, up to 30 cycles) to high numerical accuracy (certainly much higher than the processor). In fact the bond dimensions required mean a laptop could be used for this purpose. I think it would be important to make this clear to readers and even include such classical data in the main plots as it is essentially exact simulation data.

This, and the other referees comments about the fact the physics of the system can be observed without the need for 100 qubits, leads me to now have some concern about the suitability manuscript for Nature. Whilst I find it impressive the authors are able to realize these topological edge modes in a many-body system, the setup does not gain us physical insight beyond what could have been achieved via simpler, more efficient means.

Thus the main result here is the devices ability to physically realize these edge modes and their potential for quantum information storage. In my opinion, without evidence i) the device has been used to discover something unknown, ii) simulate a system outside the reach of other methods, or iii) a clearer case for the utility of these realized topological edge modes, I think the paper is not suited for Nature.

(Remarks on code availability)

Everything is appropriate. I appreciate the addition of a README.

Referee #3

(Remarks to the Author)

I have read through the revised manuscript and the authors' response to my questions. The updated manuscript is significantly improved over the initial version, with the new data substantiating the authors' claims much better. I therefore recommend the work for publication.

p.s. I still share my initial concern that the edge modes are highly localized quantities and unaffected by majority of the qubit chain, a suspicion confirmed by the authors' new edge mode profile measurements showing the weights are concentrated in just the outermost qubit. The system (away from the critical point) will therefore never be complex enough to require a quantum computer to investigate.

(Remarks on code availability)

List of major changes (marked in red in the main text and the Supplementary Information):

1. We have carried out new experiments measuring the spatial profile of the prethermal strong zero modes to address the Referees #2 and #3's concerns on the many-body nature of these modes. We have added the results to Fig. 2e as a new subfigure.
2. Following the Referee # 3's suggestion, we have carried out spectroscopy on larger systems and re-plotted Fig. 4 with the spectra of an 16-qubit system. We have added a new Extended Data Fig. 6 to display the spectra of systems with different sizes.
3. We have introduced relevant timescales in our experiments into the main text and added detailed discussions to a new Sec. 1A in the Supplementary Information.
4. We have revised the "SPT Hamiltonian and its implementation" section and Methods part "Characterization of Trotter errors" to introduce the Floquet prethermalization. We added detailed discussions to a new Sec. 1B in the Supplementary Information.
5. We have carried out new experiments to compare the noise resilience of logical qubits encoded in the edge modes versus that of physical qubits. We have added the results with detailed discussions into a new Sec. 2F in the Supplementary Information.
6. We have added a new Sec.1H in the Supplementary Information for numerical results on simulating the noiseless systems with the MPS method.
7. We have revised Sec. 1G in the Supplementary Information to discuss our improved spectroscopy method.
8. We have remeasured the data in Fig. 2b and Extended Data Fig. 3a of the main text and updated the two figures with improved data on echo circuits.
9. We have made other necessary revisions throughout the whole manuscript to correct typos, improve the presentation, and address the Referees' comments/suggestions.

Response to Referee #1

We are grateful to the Referee for their careful reading of the manuscript, precise summary of the main results, constructive suggestions, and kind recommendation. We particularly appreciate the Referee’s positive evaluation that our work “pushes the boundaries of what is possible with digital quantum simulation.” Based on their report, we have carried out additional numerical calculations to validate and extend our experimental results. We have clarified several crucial points and improved the presentation significantly. The point-by-point response to the Referee’s comments is provided below.

Comment 1 of Referee #1:

In this work the author’s report the realization of long-lived finite-temperature edge modes in a disorder-free 125-qubit quantum processor—with 100 of the qubits arranged into a one-dimensional chain. In order to achieve this, the authors implement a deep quantum circuit which realizes the discrete-time dynamics of a complex Hamiltonian with three-body interactions. The remarkably low gate errors available on the processor allow the quantum dynamics of the system to faithfully emerge even after a significant number of cycles.

The significance and import of the work is very clear: experimentally realizing finite-temperature edge states in a disorder-free system. I am unaware of experimental work realizing such states until now, and achieving this with a digital processor really pushes the boundaries of what is possible with digital quantum simulation. Due to the above, I do recommend publication of the manuscript. I do, however, have some pertinent comments about the manuscript that I think should be addressed.

Authors’ response: We sincerely thank the Referee for their positive evaluations that “the significance and importance of the work is very clear” and that our work “pushes the boundaries of what is possible with digital quantum simulation.” We greatly appreciate their recommendation for publishing our work in Nature. We value their insightful and constructive comments and have carefully addressed each of them in the following.

Comment 2 of Referee #1:

It is not clear in the main text how measurements are performed i.e., how many readout shots are taken. What does “five repetitions of measurements” mean in the context of the error bars in the figures? I presume a large number of shots were taken five times and the resulting five shot-averaged values were averaged again and the standard error on the mean was computed to get the error bars? This should be made clearer, especially as the readout error of the device is 88% and this is not reflected in the error bars – which stem solely from a very small statistical sample.

Authors’ response: We thank the Referee for raising this important point regarding the presentation. Indeed, what we did for “five repetitions of measurements” is exactly the same as described by the Referee: To estimate the expectation value of observables, we took 20,000 shots for the bulk dynamics and four-qubit tomography, and 10,000 shots for other observables. We repeated this procedure five times, averaged the resulting five shot-averaged values as the mean values, and computed their standard deviation as the error bars. The readout error of our device is about 0.88% (see Supplementary Information Fig. S8), corresponding to a readout fidelity of 99.12%. Combined with a large number of shots and the readout error mitigation (see Supplementary Information Sec. 2D), we achieved stable and accurate measurements of edge modes and stabilizers in our experiments.

Following the Referee’s suggestion, in the revised main text, we have clarified how the measurements are

performed and explicitly mentioned the number of shots in the caption of each figure.

Comment 3 of Referee #1:

I am surprised the authors have not verified/corroborated some of their results with Matrix Product State simulations of the dynamics. These should be straightforward to implement, and it would be interesting to help understand a) the growth of entanglement in the system and b) the faithfulness of the device at realizing the dynamics—at least on short/moderate timescales (depending on how fast entanglement grows and how difficult the simulations are to converge). These simulations would also help to verify the observation of an emergent $U(1)\times U(1)$ that the authors report.

Authors’ response: We thank the Referee for this constructive and enlightening suggestion, which has led us to carry out additional numerical calculations on large systems to better understand the growth of entanglements and the emergent symmetry in our system.

We first use the matrix product state (MPS) to calculate the dynamics of a small system with $N = 12$ qubits, which is accessible with exact diagonalization (ED) and yields a comparison for benchmarks. As shown in Figure R1a, MPS results (black lines) and ED results (green dots) agree precisely with each other. For the homogeneous case ($J_o/J_e = 1$, left panel in Figure R1a), we observe a rapid relaxation of the edge mode \tilde{Z}_L , as well as excitation numbers on odd sites, n_o , and even sites, n_e . As expected, the entanglement entropy S also increases rapidly and almost saturates at $t = 30$ cycles. For the dimerized case

Figure R1: MPS simulation results. **a**, Comparison between MPS and ED for a small system with $N = 12$ qubits. The initial state is a cluster state with 5 excitations. **b**, The growth of entanglement entropy and bond dimensions for three different initial states in our $N = 100$ qubit experiments. **c**, Comparison between the MPS results and experimental data for the emergent $U(1)\times U(1)$ symmetry. In both cases, the excitation numbers are obtained from the normalized expectation values of stabilizers $\langle \bar{K}_i \rangle = \langle \Psi_e | K_i(t) | \Psi_e \rangle / \langle \Psi_0 | K_i(t) | \Psi_0 \rangle$. The small deviations between the MPS and experimental data after around 15 cycles are due to accumulations of experimental imperfections. In all cases, the truncation error cutoff for MPS is set to be 10^{-6} .

($J_o/J_e = 3.17$, right panel in Figure R1a), while \tilde{Z}_L , n_o , and n_e are approximately conserved, S still grows considerably during the evolution. This growth occurs because excitations can propagate freely among the sites with the same parity, even though the dimerization suppresses edge-bulk interactions and excitation exchanges between sites with different parity (as shown in Fig. 3 of the main text).

We then use MPS to simulate the dynamics of the system with $N = 100$ qubits. In Figure R1b, we observe a nearly linear growth of entanglement entropy S with time for the dimerized case. In particular, the growth rate increases as the number of excitations in the initial state rises, which poses challenges for simulating the long-time dynamics of our system. This is in sharp contrast to the case of many-body localization, where propagation of excitations is forbidden and the entanglement grows logarithmically, rather than linearly, with time. Indeed, we find that for fixed truncation error, 10^{-6} , the bond dimensions quickly grow up to 1024 and become intractable when $t \gtrsim 30$. This aligns with the recent experimental progress (arXiv:2503.20870), which demonstrated that accurately simulating prethermal states using current classical methods is challenging. Nevertheless, we can still use the MPS results to verify our experimental data for short timescales. In Figure R1c, we compare the time dynamics of n_e, n_o between MPS results (dashed lines) and experimental data (markers). These results agree well with each other, further confirming the presence of $U(1) \times U(1)$ symmetry and the performance of our device.

In the revised manuscript, we have added a new Sec. 1H in the Supplementary Information to incorporate these numerical results. The related simulation codes are also updated to the Code Ocean repository.

Comment 4 of Referee #1:

In relation to the above, it would seem that the structure of the device means it may be capable of simulating the dynamics of a similar Hamiltonian but in two-dimensions. Can the authors comment further on the feasibility of this with their device and whether they would expect similar physics to emerge? This is briefly alluded to in the conclusion, but it is unclear whether the current device can realize such dynamics or further engineering is required.

Authors' response: We thank the Referee for raising this inspiring point. Indeed, the prethermal strong zero modes studied in our experiments are not limited to one-dimensional (1D) systems. Recent theoretical work (arXiv:2406.01686) shows that the 0D (point-like) corner modes behave as prethermal strong zero modes in a class of Hamiltonians with higher-order symmetry-protected topological phases. Such Hamiltonians can be constructed in all spatial dimensions. In 1D, it becomes the ZXZ Hamiltonian studied in our experiments, and in 2D, it is defined on the checkerboard lattice containing 5-body stabilizers. These stabilizers are local and commute with each other, and hence their evolution can be faithfully implemented via quantum circuits with local two-qubit gates.

From the experimental perspective, the square-like structure with nearest-neighbor tunable couplings of our device naturally supports digital simulation of this 2D model. Yet, currently two problems hinder us from carrying out such an experiment: 1) For one Trotter step of the 5-body stabilizer evolution, the layers of two-qubit gates are doubled compared to the ones of 3-body terms in 1D. This will shorten our experimental timescale to about 10 cycles. 2) There are some malfunctioning qubits/couplers in our device, which mainly come from the device fabrication process. In the 1D case, this can be circumvented by carefully designing geometric connections of the chain. While in 2D, implementing the 5-body stabilizers requires the functionality of all these couplers. Nevertheless, we are working hard on developing the next-generation device with improved gate fidelity and fabrication recipe. We believe our improved devices in the future will

enable the digital quantum simulation experiments for 2D systems with a large system size. We leave this interesting and important topic for future studies.

Comment 5 of Referee #1:

The data and plotting code used to produce the results is accessible and easy to run due to being well labelled and in the form of '.ipynb' with the corresponding data stored as Excel sheets. Although, there is no README file for installation/running the code, so I think a short README to describe the structure of the repository and how users less familiar with the .ipynb format can run the code would be helpful.

Authors' response: We thank the Referee for taking the time to verify our code and data. We agree with the Referee that a README file will be helpful for more general readers. Following the Referee's suggestion, we have added a detailed README file in the Code Ocean so that the readers can follow it step by step for installing/running our codes.

In summary, we greatly appreciate the Referee's positive evaluation of our work and their constructive comments/suggestions. We have carefully addressed all their comments and substantially improved the manuscript. We hope that this improved manuscript will satisfy the Referee.

Response to Referee #2

We sincerely thank the Referee for their time reviewing the manuscript and their constructive suggestions. We took the Referee's comments and suggestions very seriously. We have spent great efforts to quantitatively study the relevant time scales and carry out additional experiments to further verify the existence of prethermalization and strong zero modes. Thanks to the Referee's report, we have better clarified these important points and improved the presentation of the manuscript significantly. The following contains our detailed response to specific points.

Comment 1 of Referee #2:

In the manuscript "Observation of topological prethermal strong zero modes" the authors report on a quantum simulation in a 125 qubit quantum processor, in which they implement quantum dynamics described by a Hamiltonian that realizes symmetry protected topological states in its ground states. Such states have robust boundary modes and the authors probe this boundary mode and show that it can be made long lived depending on the choice of parameters in the dynamics. The manuscript is clearly written and enjoyable to read. It is a nice work that explores interesting topics of currently high interest.

Authors' response: We sincerely appreciate the Referee for judging that "The manuscript is clearly written and enjoyable to read" and "It is a nice work that explores interesting topics of currently high interest".

Here, we would also like to emphasize that the robust and long-lived edge mode observed in our experiment is not limited to zero-temperature states but also persists in finite-temperature states with bulk excitations. The dimerization induces an emergent symmetry that suppresses interactions between these excitations and the edge modes, allowing us to circumvent the need for strong disorder to freeze the excitations. Another contribution of our experiments is that we demonstrate that these edge modes can be used to process information at finite temperature. In particular, we encoded a Bell state within them and demonstrated its resilience against bulk excitations and symmetry-preserved noises in the dimerized regime.

Comment 2 of Referee #2:

While the work is nice to read, I find it to be rather short on quantitative detail. The theory is used mainly to tell a nice story about the data obtained from the quantum simulator but the detailed analysis of the data and discussion of theoretical consideration that would actually support the claim made in the title, namely that they actually observe topological prethermal strong zero modes, is rather minimal. I miss a detailed and honest discussion of the relevant time scales and what would really be required to experimentally observe what is claimed to observe. In its current form, the manuscript mostly nicely explains the theory, and they show nice data that shows the presence of boundary modes whose coherence time can be controlled by the parameters of the quantum circuit that is applied, but a quantitative connection between the two is missing. This gives the impression that the evidence for the prethermal strong zero modes is rather weak. For sure the manuscript needs to improve in terms of the details and to make sure that they explain those details to the reader, so as not to mislead the reader. Let me give a few comments in this regard, not necessarily in any specific order.

Authors' response: We thank the Referee for raising these crucial questions about the quantitative connection between theoretical predictions and experimental observations, which have led us to spend great effort on carrying out additional numerical simulations and experiments to further substantiate the prethermal strong zero modes (PSZMs) we observed. Below, we outline the overall picture of prethermal mechanisms

Figure R2: **a**, Timescales relevant to our experiments. **b**, the edge mode lifetime τ as a function of N and J_o/J_e , presenting two different scaling regimes $\tau \sim t_{\text{sym}}$ and $\tau \sim t_N$. **c**, In a chain with $N = 12$ qubits and J_o/J_e close to resonance points, τ is bounded by interactions with bulk excitations. As J_o/J_e deviates from the resonant points, the interaction is suppressed by the emergent $U(1) \times U(1)$ symmetry in the prethermal regime, and $\tau \sim t_{\text{sym}}$ increases. τ eventually saturates to a finite value t_N due to the hybridization between two edge modes in the finite-sized system. **d**, In the dimerized regime with $J_o/J_e = 2.618$ where edge-bulk interactions being sufficiently suppressed, $\tau \sim t_N$ scales nearly exponentially with N . All the data are obtained by noiseless simulation with Trotter evolution $U(\delta t) = U_1(\delta t)U_0(\delta t)$, effectively infinite-temperature initial state $|00 \dots 0\rangle$, and the parameters being the same as the ones in the main text ($\delta t = 0.5, J_e = \pi/5, h_x = 0.11, V_{xx} = 0.2$).

and relevant timescales in our work (Figure R2a), and provide in-depth discussions in the following reply to the Referee’s specific comments.

Our study involves two distinct prethermal mechanisms: Floquet prethermalization (governed by Trotter step $J\delta t$) and the prethermalization due to dimerization (controlled by parameters J_o/J_e). The former establishes a regime well-described by an effective time-independent Hamiltonian $H \approx H_0 + H_1$ and preserves energies until a late heating time t_* . The latter gives an emergent $U(1) \times U(1)$ symmetry with a lifetime t_{sym} , which underpins the stability of the PSZMs (see response to Comment 4). In our experiments, we primarily focus on the second mechanism, selecting $J\delta t$ such that t_* is far beyond the experimental timescales t_{exp} . Consequently, the system dynamics are effectively described by H , under which the edge mode lifetime τ is bounded by two factors: The hybridization between edge modes at each end of the chain, and the interaction between edge modes and thermal excitations in the bulk (Figure R2b). The former depends on the system size N , leading to a lifetime t_N scaling exponentially with N . The latter depends on both the effective temperature of initial states and the dimerization strength J_o/J_e . When J_o/J_e is near resonant points, thermal excitations in the bulk can interact strongly with the edge modes and result in their rapid decay. As J_o/J_e moves away from resonances, the prethermalization gives rise to the emergent $U(1) \times U(1)$ symmetry, prolonging τ to the symmetry lifetime t_{sym} (Figure R2c). In finite-sized systems, τ is ultimately upper bounded by t_N .

Taking experimental decoherence into account, the accessible time window is significantly limited ($t_{\text{exp}} \sim 30$, which is characterized by the echo experiment and zero-temperature state decay), precluding direct observation of intrinsic heating times t_* or hybridization time t_N in large systems. Nevertheless, since the edge modes and $U(1) \times U(1)$ symmetry decay more rapidly near resonances, the dependence of t_{sym} on J_o/J_e remains experimentally resolvable (see response to Comment 5).

In the revised manuscript, we have introduced these various time scales in the main text and added a new Sec. 1A in the Supplementary Information to discuss these time scales in detail.

Comment 3 of Referee #2:

Relevance of the number of qubits N and connection to relevant time scales. What role does N play in theory and experiment? One would want N to be large enough that the overlap of the boundary modes is negligible. This overlap is typically exponential in N and therefore the large number of qubits gives a times $t_{\text{boundary}} \sim \exp(N)$ that is very large such that one does not need to worry about the boundary mode overlap on the time scale of the experiment $t_{\text{exp}} \ll t_{\text{boundary}}$. Does N play any other role?

The time scale of the experiment is set by imperfections in the gates and is measure by the echo experiments for example in Fig. 2. We are forced to have $t_{\text{exp}} \sim t_{\text{echo}}$, as beyond that time we don't have coherent quantum dynamics. In the experiment $t_{\text{exp}} \sim 20$ and assuming diffusive dynamics and the fact that $\sqrt{20} \sim 5$, the quantum state spread about 10 qubits on the time scale of the experiment. Therefore when probing the boundary modes on that time scale, most of the $N = 100$ qubits are kind of irrelevant, all the dynamics on that time scale is happening at the boundary. One might guess that doing the same experiment with $N = 20$ or less would probably give the same results. And in fact, in Fig. S3 the authors compare their experiments with numerical simulations in systems with $N = 14$ and get a reasonable agreement. This is kind of clear, for example, in Fig. 3, where one can see that most of the excitations in the bulk of the system have no time to get anywhere near the boundary and are therefore irrelevant to the physics on the experimental time scale.

Authors' response: We thank the Referee for raising these important points, which helped us better clarify the role of the system size N and bulk excitations both from theoretical and experimental standpoints.

Theoretically, N influences edge mode hybridization, leading to an exponentially prolonged lifetime t_{boundary} (we denote it as t_N for simplicity), as correctly pointed out by the Referee. We verified this through numerical simulations on small systems. As shown in Figure R2d, taking $J_o/J_e = 2.618$, which is far from the first-order resonant point $J_o/J_e = 1$ for \tilde{Z}_L , we observe a near-exponential scaling with N for τ . The relationship between τ , N , and J_o/J_e shown in Figure R2b also confirms this exponential scaling for varying system sizes as the parameters deviate from resonances. This hybridization effect is also observed in our spectroscopy experiments in Fig. 4 and Extended Data Fig. 6 in the main text, where we observed a gap ζ between two edge modes, which gradually closed with increasing system sizes.

From the experimental standpoint, as restricted by the experimental timescale t_{exp} , the hybridization time t_N will not become the bottleneck of the edge mode lifetime for $N \gtrsim 20$, and the t_N for our 100-qubit system largely exceeds what is experimentally accessible. Also restricted by t_{exp} , some bulk excitations will not propagate to the edge and thus will not affect the edge modes. Therefore, the PSZMs are expected to perform similarly even with a smaller system size than in our experiments. However, we emphasize that our work extends beyond just edge behavior. A key aspect is to demonstrate the emergent $U(1) \times U(1)$ symmetry in the prethermal regime due to dimerization, where excitations deep in bulk play a crucial role. A large

system with ~ 100 qubits enables us to explore the rich bulk excitation dynamics and the distinct interacting behaviors among propagating bulk excitations, which are hard to observe in a short chain. As illustrated in Fig. 3 of the main text, the large system with an extensive bulk allowed us to observe distinct excitation dynamics across a range of system parameters J_o and J_e , improving our understanding of the dimerization mechanism. Furthermore, as mentioned, our work identifies two different prethermal mechanisms, and their coexistence in a large system over a broad energy range was not *a priori* observed. Therefore, the large system size in our experiments was essential for systematically studying these interesting phenomena.

Furthermore, we emphasize that going for 100 qubits and more is a promising route to discovering and verifying new physics. As shown in the theoretical work [Phys. Rev. B 111, L201114 (2025)], strong zero modes are not limited to the ends of chains but also emerge at the interfaces between distinct phases. Our extensive bulk allows us to engineer different phases within separate regions of the chain, leading to multiple boundary modes occurring at their interfaces. By carefully designing the system Hamiltonians, we can manipulate these boundary modes, bringing them close to each other for potential logical operations and subsequently separating them. This approach gets rid of the geometric constraints on performing logical operations using only edge modes and offers a scalable way to increase the number of strong zero modes within a single 1D chain.

Finally, we would like to highlight the significant experimental effort involved in these measurements. The quantum computation and simulation community is actively focused on scaling qubit numbers and improving control fidelity to explore exotic phases of matter. Our work exemplifies this effort: We developed a new 125-qubit superconducting processor and utilized 100 qubits with high programmability and low-error parallel gates. We stress that the programmability and low error rates of our platform enable the exploration of extended models exhibiting similar prethermal mechanisms. For instance, the model studied here can be extended to two dimensions to explore higher-order SPT phases and corner modes, which can be readily implemented on our device with a considerably large system size.

Comment 4 of Referee #2:

Usually I think of prethermalization to happen in driven systems. Driven systems tend to heat up to infinite temperature and only when there are some approximate conservation laws one can obtain an exponentially large time scale t_{pre} before which the system reaches a different prethermal state and only after much larger time scale does it thermalize to infinite temperature. Most of the discussion in the paper however is in terms of Hamiltonian dynamics which, since it conserves energy, does not have the same infinite temperature phenomena. Now in the experiment the authors don't actually implement the Hamiltonian dynamics but instead Trotterize the dynamics and through this actually effectively get a driven system. So the physics of prethermalization enters through this Trotterization. This is not clearly explained in the manuscript and the discussion of the physics of prethermal states could be more clearly explained and the relevant time scales explained and given explicitly. Some of this can be kind of obtained from the supplemental material, but even there they authors are not careful in clearly separating the discussion of the Hamiltonian dynamics and the driven system dynamics and what facts belong to which case.

Authors' response: We thank the Referee for raising this insightful comment, which has helped us clarify further the prethermal mechanisms involved in our model and explain the relevant time scales in the revised manuscript.

We agree with the Referee that the Trotterized evolutions will give rise to Floquet prethermalization, leading

to an effective Hamiltonian $H = H_0 + H_1 + O(J\delta t)$ and almost conserved energies until a heating time t_* that is exponential in $1/(J\delta t)$ [for example, see Phys. Rev. Lett. 115, 256803 (2015); Phys. Rev. Lett. 116, 120401 (2016); Commun. Math. Phys. 354, 809 (2017)]. However, this prethermal mechanism does not give rise to the emergent $U(1)\times U(1)$ symmetry observed in our experiments, and hence is not the origin of the robust edge modes. We verify this by numerical simulations and show the results in Figure R3a. It is clear that the system heating rate measured by $|E(t)/E(t=0)|$ is slowed down with decreased Trotter step size, while the lifetime of neither edge modes nor $U(1)\times U(1)$ symmetry is enlarged. Therefore, the emergent $U(1)\times U(1)$ symmetry and long-lived edge modes do not originate from Floquet prethermalization due to Trotterization.

Instead, the emergent $U(1)\times U(1)$ symmetry originates from the structure of the effective static Hamiltonian H : in particular, from the integer number spectrum of the unperturbed cluster Hamiltonian when its stabilizer strengths are away from the resonant points [for example, see Phy. Rev. X. 7, 011026 (2017); Phy. Rev. X. 10, 021032 (2020)]. If the system parameters are dimerized to be off-resonant, the system will preserve an approximate $U(1)\times U(1)$ symmetry up to a time t_{sym} exponential in the ratio of the largest energy scale of H_0 compared to the perturbation H_1 . Note that this only depends on the parameters of the effective static Hamiltonian, and not on the Trotterization time step, δt . Thus, after a short local equilibrium time, the system enters a prethermal state with both conserved energy and $U(1)\times U(1)$ symmetry. This aligns with our numerical results: In Figure R3b, we observe that both $U(1)$ symmetries on the odd and even sites become approximately conserved as J_o/J_e deviates from the first-order resonant point from 1 up to 3.17, leading to prolonged lifetime of the edge modes. Crucially, further increasing J_o to 4.618 will lead to the breakdown of Floquet prethermalization, resulting in rapid thermalization and again short lifetimes of edge mode and symmetry (dark curve). In our experiments, we bounded the parameter J_o/J_e up to 3.2 to avoid

Figure R3: Prethermalization via Floquet evolutions (a) and dimerization (b). **a**, Short-step Trotterized evolutions induce the prethermal regime with effective Hamiltonian and almost conserved energy (bottom panel, light curves), while the edge modes and $U(1)\times U(1)$ symmetry are not conserved. **b**, The deviation of J_o/J_e from the resonant point induces the prethermal regime with the emergent $U(1)\times U(1)$ symmetry and long-lived edge modes. The data is calculated with $N = 14$ and the initial state being the cluster state with 4 excitations.

this rapid heating.

Following the Referee’s suggestion, in the revised manuscript we have clarified that our work focuses on the prethermalization due to dimerization instead of Floquet prethermalization. We added a new Sec. 1B in the Supplementary Information to discuss the Floquet prethermalization and revised Sec. 1E to support our discussion about prethermalization due to dimerization with numerical results.

Comment 5 of Referee #2:

Once such a discussion is given and the relevant time scales are given, I suspect that $t_{\text{pre}} \gg t_{\text{exp}}$. Which results in the question of in what sense do the authors actually see prethermalization? There is no plateauing of expectation values of boundary modes that one could interpret as the system reaching a prethermal state. Instead one simply sees decay. This needs to be discussed and explained in the main text.

Authors’ response: We thank the Referee for this important question. We agree that the prethermalization lifetime, t_{sym} , significantly exceeds the experimental timescale, t_{exp} , when the parameter J_o/J_e is far from the resonant points. However, near resonance, t_{sym} is significantly shorter ($\sim 10^1$), making it experimentally feasible to observe the presence and absence of prethermalization by tuning J_o/J_e . This is evidenced by three key experimental observations as J_o/J_e deviates from resonances:

1. As shown in Fig. 2d of the main text, starting from the finite-temperature state $|\Psi_e\rangle$, the edge mode lifetimes τ are largely prolonged.
2. In Fig. 3a-c of the main text, the excitation dynamics transition from diffusive propagation to motion restricted within sites of the same parity.
3. In Fig. 3d-f of the main text, the normalized excitation numbers on even and odd sites, n_o and n_e , become approximately conserved and form plateaus within t_{exp} .

In the revised manuscript, we have clarified these points and provided a more in-depth explanation of how our experimental results support the observation of prethermalization due to dimerization in the main text and Supplementary Information.

Comment 6 of Referee #2:

Related to the last point, since $t_{\text{exp}} \ll t_{\text{pre}}, t_{\text{boundary}}$, in which sense does the experiment actually demonstrate symmetry protected topological state?

Authors’ response: We thank the Referee for raising this point. Our study simulates the evolution governed by the cluster Hamiltonian H_0 with perturbation terms H_1 . The evolution begins with certain cluster states, which are eigenstates of H_0 . We measured the bulk stabilizers K_i of these initial states and showed each $|K_i|$ is very close to 1 in Extended Data Fig. 2, indicating that they are close to the SPT states.

A key characteristic of SPT states is their robustness against local, symmetry-preserving perturbations. In our experiment, these perturbations include the $h_x \sigma_i^x$ and $V_{xx} \sigma_i^x \sigma_{i+1}^x$ terms in H_1 . Following the Referee’s suggestion, we have conducted additional experiments to highlight the resilience of the edge modes to symmetry-preserving perturbations, which is in sharp contrast with the unprotected physical qubits. The experimental circuits for this comparison are illustrated in Figure R4a, where edge modes evolve under the cluster Hamiltonian H_0 , and physical qubits are subjected to Rabi π pulses. In both scenarios, we introduced a layer of R_x rotation gates to emulate symmetry-preserving perturbations with tunable strength h_x .

Figure R4: The robustness of SPT states against symmetry-protected perturbations. **a**, Experimental circuits for time evolution of SPT states (top panel) and physical qubits (bottom panel), with execution time per cycle equal to 200 ns for both cases. For SPT states, we set $J_o = 3.17J_e = 3.17\pi/5$ and consider the edge modes as logical qubits. **b**, Measured dynamics of edge modes and physical operators with ($h_x = 0.11$, or random h_x chosen uniformly from $[-0.11, 0.11]$) and without ($h_x = 0$) perturbations. **c**, Fidelity dynamics of logical and physical Bell states. Error bars in **b** and **c** represent the standard deviation (for $h_x = 0$ and $h_x = 0.11$) or the standard error of the statistical mean (for random h_x) over five rounds of measurements, with each round taking 10,000 shots for logical qubits and 3,000 shots for physical qubits.

Figure R4b presents the lifetime of logical and physical Z and Y operators. At $h_x = 0$, the edge modes exhibit a faster decay compared to physical qubits, primarily due to errors from the additional two-qubit gates involved. However, as h_x increases to 0.11, the physical operators show significant oscillations, whereas the edge modes remain largely preserved. Similar behavior is observed for the Bell state fidelity, as depicted in Figure R4c. These results clearly demonstrate that the states prepared in our experiments exhibit robustness against symmetry-preserving perturbations, which is a direct consequence of their SPT nature.

In the revised manuscript, we have highlighted this robustness against symmetry-preserving perturbations in the main text and added a new Sec. 2F in the Supplementary Information for experimental results and in-depth discussions.

Comment 7 of Referee #2:

How does the data give evidence for the presence of a strong zero mode, instead of simply a boundary mode? I'm missing quantitative details in this regard.

Authors' response: We thank the Referee for raising this critical question, which has led us to perform

additional experiments to quantitatively characterize the prethermal strong zero mode (PSZM).

Ordinary boundary modes are protected with the restriction to the states at the bottom or top of the spectrum, because their spectral degeneracy is only robust within these zero-temperature manifolds. In sharp contrast, PSZMs can survive finite temperatures. By definition, PSZMs are operators that approximately commute with the system Hamiltonian with exponentially small corrections while anticommute with specific symmetry operators [for example, see *J. Phys. A: Math. Theor.* 49, 30LT01 (2016); *J. Stat. Mech.* 2017, 063105 (2017); *Phys. Rev. X* 7, 041062 (2017)]. This extends the robust zero-temperature manifold degeneracy across the entire spectrum, thereby enabling robust and long-lived edge operators for arbitrary initial states. In our system, two conjugate PSZMs, Ψ^z and Ψ^x , localize at each end of the chain. For the cluster Hamiltonian H_0 with zero correlation length, these PSZMs are explicitly represented by the edge operators $\sigma_1^z, \sigma_1^x \sigma_2^z$ on the left, and $\sigma_N^z, \sigma_N^x \sigma_{N-1}^z$ on the right. In the presence of perturbation H_1 , PSZMs would extend into the bulk. We provided the first-order descriptions for Ψ_L^z and Ψ_L^x in Eqs. (S8–S9) of the Supplementary Information. Here, for convenience we also show these equations in the following:

$$\begin{aligned}\Psi_L^z &= \tilde{Z}_L + \frac{h_x}{J_e} \sigma_1^x \sigma_2^x \sigma_3^z - \frac{V_{xx}}{J_o^2 - J_e^2} (J_e \sigma_1^x \sigma_3^z + J_o \sigma_1^y \sigma_2^y \sigma_3^x \sigma_4^z), \\ \Psi_L^x &= \tilde{X}_L + \frac{h_x}{J_o} \sigma_1^x \sigma_2^x \sigma_3^x \sigma_4^z + \frac{V_{xx}}{J_o^2 - J_e^2} (J_o \sigma_2^x \sigma_3^x \sigma_4^z + J_e \sigma_1^z \sigma_2^z \sigma_3^z) \\ &\quad - \frac{V_{xx} J_e}{J_o^2 - 4J_e^2} \left[\sigma_1^y \sigma_2^z \sigma_3^y + \left(\frac{2J_e}{J_o} - \frac{J_o}{J_e} \right) \sigma_1^x \sigma_2^x \sigma_4^z - \sigma_1^x \sigma_2^y \sigma_3^y \sigma_4^x \sigma_5^z - \frac{2J_e}{J_o} \sigma_1^y \sigma_4^y \sigma_5^z \right].\end{aligned}\quad (1)$$

Figure R5: The spatial profile of the prethermal strong zero modes Ψ_L^z (a) and Ψ_L^x (b) on the left end of the chain. The solid boxes denote theoretical predictions derived from Eq. (1), where black and red frames mark positive and negative values, respectively. In each case, the coefficients are obtained by averaging the late-time dynamics over cycles from $t = 25$ to 40, with the sum of their squares normalized to 1.

A characterization of PSZMs, which can be found in these equations, is the divergence at resonant points, indicating PSZMs extend to the bulk and become highly non-local. In the original manuscript, we observed that there are dips in the edge operator lifetime at $J_o/J_e = 1$ and 2 in Fig. 2d, which demonstrates the divergence in Ψ_L^z , Ψ_L^x , and hence differs PSZMs from ordinary boundary modes.

Based on the Referee's comments, we have conducted additional experiments to measure the coefficients of each Pauli operator in Ψ_L^z , Ψ_L^x as their spatial profiles. We followed the method introduced in Ref. [39], measuring the late-time dynamics of each Pauli operator, which gives an amplitude proportional to the coefficient. These results, now included in revised Fig. 2 of the main text (and shown in Figure R5), reveal two important points: 1) for Ψ_L^z (Figure R5a), decreasing J_o/J_e from 3.17 to 1.5 (approaching the resonant point $J_o/J_e = 1$), reduces the σ_1^z edge coefficient while enhancing bulk contributions, signaling the delocalization of Ψ_L^z ; 2) as J_o/J_e decreases further to 0.6, sign reversals occur in the coefficients of $\sigma_1^x \sigma_3^z$ and $\sigma_1^y \sigma_2^y \sigma_3^x \sigma_4^z$, indicating J_o/J_e across a resonant point. We observe similar behavior for Ψ_L^x (Figure R5b), where sign reversals arise twice due to low-order resonances at $J_o/J_e = 1$ and 2. These results demonstrate that the edge modes observed in our experiment are PSZMs, rather than simply boundary modes.

Comment 8 of Referee #2:

So, in summary, I feel the main text of the manuscript needs to be a bit more quantitative in analysing all relevant time scales and if they want to argue that what they observe is actually strong prethermal zero modes they need to argue how prethermalization arises in this system and give evidence that they have reached the prethermal state. Some motivation for how one can experimentally verify the presence of strong zero modes versus just a boundary state would also help improve the paper. Of course, the theoretical arguments for why one should observe a prethermal strong zero modes in this system is pretty robust and well known. While the paper describes this theory nicely (apart from some details as discussed above) there is in principle nothing new in terms of the physics here. My main concern with the manuscript is that it's not clear to me that this physics is accessible on the time scale allowed by the experiment.

Authors' response: We greatly appreciate the Referee's valuable comments and suggestions, which are very helpful for us to improve the manuscript. As discussed in-depth above, in this revised manuscript we have quantitatively studied relevant time scales and verified the presence and absence of prethermalization by tuning the system parameters. We also carried out additional experiments to measure the spatial profiles of PSZMs and hence verified that they are distinct from a simple boundary state. Our work reports the first experimental demonstration of the PSZMs in a disorder-free system, observing the divergences near resonances and emergent $U(1) \times U(1)$ symmetry away from resonances, and demonstrating their potential application in robust information processing at finite temperatures.

We have carefully addressed all the important points raised by the Referee and significantly improved the presentation. We hope that the improved manuscript will satisfy the Referee and convince them to recommend publication of this work in Nature.

Response to Referee #3

We are grateful to the Referee for their time reviewing the manuscript, accurate summary of the main results, and constructive suggestions. The Referee has provided a carefully written report, which was invaluable for us to improve the paper. We took the Referee's comments and suggestions very seriously and have spent great efforts on carrying out additional experiments to address these comments/suggestions. Based on the Referee's report, we have clarified several important points and improved the presentation of the manuscript significantly. The following contains our detailed response to specific points.

Comment 1 of Referee #3:

This manuscript "Observation of topological prethermal strong zero modes" by Jin et al. presents an experimental study of a stabilizer Hamiltonian which can be mapped to two coupled Majorana chains. The four-fold degeneracy of the ground state gives rise to two coupled edge modes on each end of the chain. The authors prepare bulk excitations by flipping some of the stabilizers away from the edges, propagating the excitations using an interaction term, and measuring edge modes over time. For specific choices of model parameters, the authors report that the edge modes are largely unaffected by the bulk excitations due to a prethermalization mechanism. Lastly, the authors prepared Bell states using the edge modes and show that they also decay slowly under the same choice of model parameters.

The possibility of long-lived edge modes in stabilizer models, and particularly the potential of using them as qubits, is of interest to the quantum simulation and quantum information community. However, since much of the theoretical finding of this work has already been published [e.g. Ref. 14], my evaluation will apply primarily to the experimental demonstration of the edge mode effect. To this end, I find the current manuscript rather lacking in terms of depth. The figures are largely demonstrations of known theoretical results using already established experimental techniques (spectroscopy, echo etc). There is little effort to illustrate any unique insights coming from performing these tasks on an experimental quantum processor. Let me elaborate:

Authors' response: We thank the Referee for the nice summary of our results. We appreciate the Referee for acknowledging that the physics explored in our manuscript "is of interest to the quantum simulation and quantum information community," which is also pointed out by Referee #2 that "It is a nice work that explores interesting topics of currently high interest." We would like to emphasize that our work provides the first experimental observation and characterization of prethermal strong zero modes (PSZMs). The direct observation of resonances in edge mode lifetime, bulk excitation dynamics, and bulk-edge interactions requires faithfully implementing the evolution of the interacting Hamiltonian without disorder, which is not trivial and has not been reported yet as far as we know. Therefore, we believe that our work indeed represents a significant step in studying long-lived boundary modes at finite temperature and would have broad impact, which is also acknowledged by Referee #1 as "the significance and import of the work is very clear."

We agree with the Referee that PSZMs have been previously explored theoretically. However, we emphasize that the model investigated in our experiments is fundamentally different: prior works [e.g., Phys. Rev. X 7, 041062 (2017); Phys. Rev. Lett. 125, 200506 (2020)] primarily focused on time-independent Hamiltonians, whereas our experiment implements a Trotterized evolution. This time-dependent process can potentially introduce more resonances in energy absorption within the system, and the robustness of PSZMs under such Trotterized evolution is not immediately obvious. Our experiment validates this robustness, which we find can be attributed to another prethermalization mechanism: Floquet prethermalization (see response to Com-

ment 4 of Referee #2). Therefore, our work extends beyond a mere demonstration of existing theoretical results, offering a systematic experimental investigation of prethermalizations and PSZMs.

In addition, we would like to highlight the significant experimental efforts required to perform these experiments. Indeed, in order to carry out these experiments we have developed a new 125-qubit superconducting processor and achieved high-fidelity parallel quantum gates across a large number of qubits, which requests tremendous efforts. The availability of such large-scale, high-fidelity quantum processors opens up exciting avenues for future research, such as manipulating multiple boundary modes within 1D chains and observing robust corner modes in 2D models that we discuss in the following.

To address the Referee’s comments and suggestions, we have carried out additional experiments on measuring the spatial profile of edge modes, comparing the performance of Bell states encoded with PSZMs and physical qubits, and measuring the spectra for larger systems with an improved spectroscopy method.

Comment 2 of Referee #3:

The system size of 100 qubits in this study seems impressive at first sight. It is disappointing to then see that the experiment does little to utilize this system size, or even illustrates that the edge modes are truly many-body in nature. For instance, there is no measurement of spatial profile of the edge modes to show that they in fact extend beyond the edge qubits under the authors’ choice of model parameters. If the spatial profile does not extend beyond more than a few qubits, then it begs the question of whether having such a large system size is truly meaningful and if the experimental data will look effectively identical with just a dozen qubits.

Authors’ response: We appreciate the referee’s candidness with respect to the role of system size in our work; we will return to this point in detail in our next comment. However, first, we would like to address the referee’s concern with regards to our analysis of the edge modes. Theoretically, the PSZMs in our model are many-body operators that approximately commute with the system Hamiltonian, with small high-order corrections. The first-order descriptions of the PSZMs Ψ_L^z, Ψ_L^x on the left end of the chain are provided in Eqs. (S8–S9) of the Supplementary Information. Here, for convenience we also show these equations in the following:

$$\begin{aligned}\Psi_L^z &= \tilde{Z}_L + \frac{\hbar_x}{J_e} \sigma_1^x \sigma_2^x \sigma_3^z - \frac{V_{xx}}{J_o^2 - J_e^2} (J_e \sigma_1^x \sigma_3^z + J_o \sigma_1^y \sigma_2^y \sigma_3^x \sigma_4^z), \\ \Psi_L^x &= \tilde{X}_L + \frac{\hbar_x}{J_o} \sigma_1^x \sigma_2^x \sigma_3^x \sigma_4^z + \frac{V_{xx}}{J_o^2 - J_e^2} (J_o \sigma_2^x \sigma_3^x \sigma_4^z + J_e \sigma_1^z \sigma_2^z \sigma_3^z) \\ &\quad - \frac{V_{xx} J_e}{J_o^2 - 4J_e^2} \left[\sigma_1^y \sigma_2^z \sigma_3^y + \left(\frac{2J_e}{J_o} - \frac{J_o}{J_e} \right) \sigma_1^x \sigma_2^x \sigma_4^z - \sigma_1^x \sigma_2^y \sigma_3^y \sigma_4^x \sigma_5^z - \frac{2J_e}{J_o} \sigma_1^y \sigma_4^y \sigma_5^z \right].\end{aligned}\tag{2}$$

In particular, the coefficients of bulk terms diverge when J_o/J_e reaches certain values, which are called resonant points. Around these resonant points, the PSZMs extend to the bulk, resulting in the short edge-mode lifetime as we observed in Fig. 2d of the main text. These results demonstrate that the PSZMs in our experiments are not single-body physics.

Following the Referee’s comments, we have conducted additional experiments to characterize the spatial profiles of Ψ_L^z, Ψ_L^x , with results now incorporated in the revised main text Fig. 2 (and also shown in Figure R6). For Ψ_L^z , we observed that as J_o/J_e approaches the low-order resonant point $J_o/J_e = 1$, the coefficient of edge operators σ_1^z gradually decreases and the coefficients of the bulk terms increase. This

Figure R6: The spatial profile of the prethermal strong zero modes Ψ_L^z (a) and Ψ_L^x (b) on the left end of the chain. The solid boxes denote theoretical predictions derived from Eq. (2), where black and red frames mark positive and negative values, respectively. In each case, the coefficients are obtained by averaging the late-time dynamics over cycles from $t = 25$ to 40, with the sum of their squares normalized to 1.

verifies that the PSZMs indeed extend into the bulk, confirming their many-body nature. As J_0/J_e further decreases to 0.6, the signs of coefficients for $\sigma_1^x \sigma_3^z$ and $\sigma_1^y \sigma_2^y \sigma_3^x \sigma_4^z$ reverse, indicating the divergence of bulk terms as the system crosses a resonant point. Similar behavior is observed for Ψ_L^x , where sign reversals now arise twice due to low-order resonances at $J_0/J_e = 1$ and 2. These sign reversals and resonant effects have never been observed in previous related works such as Refs. [37-39], therefore represent unique features of the PSZMs and the emergent $U(1) \times U(1)$ symmetry in our experiments.

Comment 3 of Referee #3:

The time dynamics for all of the figures are very short (20 to 30 cycles). Given the local nature of the Hamiltonian, the lightcone of 40 to 60 qubits in the middle of the chain does not even cover the edge qubits and therefore has zero impact whatsoever. This can be easily seen in the experimental data shown in Fig. 3, where the bulk excitations between sites 25 and 75 simply do not have enough time to spread to the edges. This again begs the question whether the large system size used in this experiment is truly meaningful, and whether the experimental findings are effectively reproducible with a very small system size that one can easily study with a laptop.

Authors' response: We thank the referee for raising this very important question regarding the interplay between the system size and the finite experimentally accessible quantum simulation timescales. We will try our very best to answer the referee's question from two complementary perspectives: (i) first, specifically in the context of strong zero modes, how does having such a large system size benefit us? and (ii) from a broader perspective, given the current experimental coherence times, is having a system size of ~ 100 qubits

completely overkill or not? For the former question, we want to be very honest with the referee and say that, although having a large system size certainly helps to make the experimental interpretation cleaner in many regards, we believe that there is no fundamental qualitative feature of the strong-zero-mode-physics that is *uniquely* enabled by our ~ 100 qubit system size given the accessible timescales.

(i) Let us begin by saying that we completely agree with the referee that the experimental timescales are insufficient for all of the spatial regions of the system to be connected by the dynamics' light-cone. That being said, in addition to highlighting the experimental capabilities, we do believe that having access to such a large system size was quite important for us to be able to make clear interpretations about the experimental signatures of strong-zero modes; we will emphasize three such points below.

First, one of the key challenges in experimentally probing edge modes (whether strong zero modes or otherwise) in general is ensuring that there is a sufficiently clear distinction between the edge and the bulk. For topological edge modes, this distinction usually requires one to have a system size that is much larger than the localization length of the edge mode. For strong-zero modes (or approximate strong zero modes), this distinction is even more subtle, since the relevant localization can diverge as one nears a pole. In our case, having a relatively large chain allows us to clearly distinguish between bulk and edge excitations (although one might argue that for this specific goal, having a system size of $N \sim 50$ would also have been enough), while also enabling us to directly explore bulk excitation dynamics, and in particular, to see the distinct interacting behaviors among propagating bulk excitations. This last point is one that we will expound upon a bit in part (ii) of the response, but we will also go into a bit of detail here. In particular, as we mentioned before, the referee is totally correct that excitations in the middle of the chain do not have sufficient time to spread to the edge. In some sense, this type of “indicator” is one that is perhaps a bit motivated by the goal of generating entanglement across the full system. Let us respectfully imagine asking a different, more quantum-dynamics-type question: What are the dynamics of interacting bulk excitations in a strong-zero-mode Hamiltonian? Perhaps, one of the most natural ways to answer this question is to explore the dynamics of bulk excitations in the middle ~ 20 - 30 sites of a large system, as we have done. To answer this type of dynamical question, having a system size of ~ 100 qubits, even if one is only probing correlation functions of the middle ~ 20 - 30 sites is crucial to ensure that finite size effects do not come into play! Indeed, zooming out a bit further, this type of scenario and perspective is not unique to our discussion above. For experiments related to the emergence of hydrodynamics in quantum models, one often only attempts to probe the “middle” of a system, in order to ensure that the dynamics are not affected by finite size effects associated with excitations bouncing off a boundary.

Second, as alluded to in passing above, having access to a larger system size enables us to perform a more fine-grained study of the mechanisms underlying the stability of the edge modes. For example, a key aspect of our experiment is to demonstrate the emergent $U(1) \times U(1)$ symmetry in the prethermal regime due to dimerization, where excitations deep in the bulk play a crucial role. As illustrated in Fig. 3 of the main text, our large system, which exhibits a relatively extended bulk, allowed us to observe distinct bulk excitation dynamics across varying system parameters J_o/J_e ; this observation allowed us to significantly improve our understanding of the dimerization and the underlying prethermal mechanisms.

Finally, the relatively large size of our 1D chain allows us to further characterize the stability and robustness of our observed dynamics, with respect to many-body effects arising from interactions. Indeed, as the referee is well aware, in periodically driven systems, the late-time equilibrium state always moves toward an infinite temperature state. At larger system sizes, there are a greater number of relevant interaction terms that can

facilitate the dynamics toward this late-time featureless state. The fact that our experiment observes not one, but two distinct pre-thermalization mechanisms at system sizes never explored before, offers a verification of nontrivial theoretical predictions on the thermalization of large-scale quantum systems. Such benchmarks are important to clarify and test our understanding of the propagation (or not) of quantum information in large-scale quantum systems, and also act as a powerful benchmark of our experimental platform itself.

(ii) Zooming out a bit, although we already touched on a few points regarding quantum dynamics, we would like to mention two other directions in the context of strong-zero-modes, where our ~ 100 qubit system size and timescales would be ideal for exploring new physics. First, as shown in recent theoretical work [Phys. Rev. B 111, L201114 (2025)], strong zero modes are not necessarily limited to the ends of a 1D system but can also emerge at the interface between distinct phases. This is a lesson that seems quite intuitive from our knowledge of topological order. But once again, the generalization of this lesson to strong zero modes is more subtle — indeed, in certain cases, even for the boundary between two distinct phases, strong zero modes are unstable, while in other cases, even the boundary between two regions in the same phase, can host an approximate strong zero model. The relatively large and extended bulk demonstrated in our current manuscript allows us to engineer different phases within separate regions of the chain, leading to multiple boundary modes occurring at their interfaces. Beyond the perspective of just exploring these boundary strong zero modes, from a more operational, quantum information perspective, by carefully designing the system Hamiltonians, we can manipulate the position of these boundary modes, bringing them close to each other only when attempting to generate interactions between separate boundary strong-zero modes. Second, moving beyond one dimension, the square-like structure and nearest-neighbor tunable couplings of our device naturally support the digital simulation of 2D models with their corner modes being PSZMs (see arXiv:2406.01686). Exploring prethermalization and emergent symmetries in this 2D system may give rise to new phenomena manifest beyond 1D chains.

Comment 4 of Referee #3:

A main practical selling point, which the authors hint at using Fig. 5, is the usage of edge modes as a quantum memory. However, the manuscript does not explore at all why this quantum memory is advantageous to a single uncoupled physical qubit. What does this complicated many-body model buy us in terms of noise protection? Is there a noise direction to which the edge modes are insensitive but physical qubits are? Can the authors demonstrate this advantage in experiment?

Relatedly, there is no exploration or discussion of the role of extrinsic decoherence and whether that affects the edge modes differently compared to a physical qubit. For example, how does the Bell state decay rate in Fig. 5 compare with what you could achieve with just those two edge qubits alone without the other 98 qubits? If it's not any better, then why would we adopt such a convoluted scheme?

Authors' response: Theoretically, the edge modes of the SPT Hamiltonian are robust against local, symmetry-preserving perturbations (e.g., the $h_x \sigma_i^x$ and $V_{xx} \sigma_i^x \sigma_{i+1}^x$ terms considered in our experiment). While these perturbations can induce hybridization of edge modes, the resulting lifetime t_N is predicted to be finite but exhibits exponential scaling with the system size N . In contrast, unprotected physical qubits are vulnerable: a constant local single-body σ_i^x perturbation directly induces oscillations, leading to an immediate loss of encoded information. Similarly, unexpected two-body $\sigma_i^x \sigma_{i+1}^x$ couplings between physical qubits could also cause rapid decoherence.

Based on the Referee's comments, we have performed additional experiments to show the resilience of

Figure R7: Comparison of noise resilience between logical and physical qubits. **a**, Experimental circuits for time evolution of logical (top panel) and physical (bottom panel) qubits, with execution time per cycle equal to 200 ns for both cases. For logical qubits, we set $J_0 = 3.17J_e = 3.17\pi/5$. The initial states of logical (physical) qubits are prepared in either $|\tilde{0}\rangle_L |\tilde{0}\rangle_R$ ($|00\rangle$) for measuring $\langle \tilde{Z} \rangle$ ($\langle Z \rangle$), $(|\tilde{0}\rangle_L + i|\tilde{1}\rangle_L) \otimes (|\tilde{0}\rangle_R + i|\tilde{1}\rangle_R)$ ($(|0\rangle + i|1\rangle) \otimes (|0\rangle + i|1\rangle)$) for measuring $\langle \tilde{Y} \rangle$ ($\langle Y \rangle$), or $|\tilde{0}\rangle_L |\tilde{0}\rangle_R + i|\tilde{1}\rangle_L |\tilde{1}\rangle_R$ ($|00\rangle + i|11\rangle$) for Bell state fidelity. **b**, Measured dynamics of logical and physical operators with ($h_x = 0.11$, or random h_x chosen uniformly from $[-0.11, 0.11]$) and without ($h_x = 0$) perturbations. **c**, Fidelity dynamics of logical and physical Bell states. Error bars in **b** and **c** represent the standard deviation (for $h_x = 0$ and $h_x = 0.11$) or the standard error of the statistical mean (for random h_x) over five rounds of measurements, with each round taking 10,000 shots for logical qubits and 3,000 shots for physical qubits.

the edge modes to symmetry-preserving noises, in stark contrast to unprotected physical qubits. The experimental circuits are depicted in Figure R7a, where edge modes evolve under the SPT Hamiltonian H_0 and the physical qubits are subjected to Rabi π pulses. In each scenario, we introduced a layer of R_x rotation gates to mimic symmetry-preserving noise with tunable strength h_x . Figure R7b shows the lifetime of logical and physical Z, Y operators. With $h_x = 0$, the edge modes decay more rapidly than the physical qubits, primarily due to errors from the additional two-qubit gates involved. As h_x increases to 0.11, the physical operators exhibit pronounced oscillations, whereas the edge modes are almost unchanged. Similar behavior is observed for the Bell state fidelity shown in Figure R7c. From an experimental perspective, such perturbations can be modeled as low-frequency noise, where the noise strength remains approximately constant within a single experimental run but fluctuates across runs. To simulate this, we randomly sampled 5 sets of h_x from $[-0.11, 0.11]$ to represent the strength of low-frequency noise in each run, with the averaged values over these runs also displayed in Figure R7b, c. Under this noise model, the physical qubits decay significantly faster than the edge modes, clearly demonstrating that the edge modes are more resilient to low-frequency noises than physical qubits.

We emphasize that the primary goal of our work is not to demonstrate immediate performance “break-even” where the edge modes necessarily outperform physical qubits under all conditions in the current experimental setup. Instead, we aim to establish these SPT edge modes as promising candidates for noise-resilient qubits for encoding and manipulating quantum information at finite temperature. While prior studies (Refs. [37-38]) have shown that edge modes in SPT chains can exhibit long lifetimes under specific conditions like many-body localization or quasi-periodic driving, our study represents the first experimental realization of logical operations and entanglement encoding using these edge modes. This also distinguishes our work from the noise-resilient edge mode discussed in Ref. [39], where only a single non-locally encoded qubit at the chain ends was considered, and performing logical operations on it could be significantly more challenging. More importantly, we explicitly demonstrate that the prethermal mechanism can protect quantum information and entanglement encoded in these edge modes against excitations. This finding opens new avenues for the potential applications of these protected edge modes in future quantum information processing.

In the revised manuscript, we have mentioned the noise resilience of edge modes against symmetry-preserving perturbations in the main text and added a new Sec. 2F in the Supplementary Information for experimental results and detailed discussions.

Comment 5 of Referee #3:

Without addressing these fundamental questions, I feel that the manuscript will not deliver the impact that a quantum simulation work in Nature should aim at delivering, especially given the preponderance of closely related edge mode papers that have already been published (e.g. Refs 37 and 39).

Authors’ response: We thank the Referee for this important point regarding the potential impact of our manuscript, particularly in light of previous related edge mode research. We agree with the Referee that establishing clear novelty and impact beyond existing work is essential for a Nature publication. Our work tackles a fundamentally distinct experimental scenario: the observation and characterization of prethermal strong zero modes in a *disorder-free*, interacting system. This stands in stark contrast to Refs. [37-39], all of which incorporate disorder in their models. A defining feature of our disorder-free, interacting model is that, unlike the nearly localized excitations typically found in disordered systems, which hardly interact with the bulk, our system exhibits pronounced bulk excitation dynamics, enabling experimental investigation of their propagation and interaction with edge modes.

Furthermore, we report the first experimental observation of resonance phenomena in prethermal strong zero modes. This behavior arises from the emergent $U(1)\times U(1)$ symmetries in the prethermal regime, a feature absent and unreported in previous studies such as Ref. [39]. Besides, our work also represents a step towards leveraging these modes for quantum information processing—performing logical operations and encoding entanglement at finite temperature. These experimental findings—spanning from observing novel physical phenomena and bulk-edge interactions in a disorder-free system to demonstrating foundational logical operations and entanglement—directly address the timely questions about finite-temperature long-lived edge modes in clean systems and their potential utility, thereby differentiating our work from earlier studies.

Comment 6 of Referee #3:

Some other more specific questions:

1. Within the introduction of the manuscript, the authors wrote “This enables us to successfully implement

the dynamics of a prototypical SPT Hamiltonian (Fig. 1b) in different regimes with quantum circuits of depth exceeding 270 and gate counts above 17, 000...” This is highly misleading. The majority of the gates do not contribute at all due to the short depth and limited lightcones of the circuits, and the rest of the gates have little weight on the observables due to the slow spreading of excitations (and therefore noise as well).

Authors’ response: We thank the Referee for raising this point. We agree with the Referee that stating the total circuit depth and gate count at this stage without further contextual explanation could potentially be misinterpreted as implying that all gates contribute equally to the edge mode behavior. In the revised manuscript, we have substituted this sentence with “This enables us to successfully implement the dynamics of a prototypical SPT Hamiltonian (Fig. 1b) in different regimes.” without mentioning circuit depth and gate number.

However, we would like to clarify that the full circuit implementation, including the gates in the bulk, is indeed essential for our study, as our work does not only focus on the edge modes but also on investigating the bulk dynamics associated with the emergent $U(1)\times U(1)$ symmetry. The gates throughout the bulk are necessary for generating the specific patterns of excitation dynamics observed in different parameter regimes. Studying these dynamics, which involve the evolution and interaction of excitations within the bulk, is crucial to unraveling the prethermal mechanism in our model.

Comment 7 of Referee #3:

2. Fig. 2b: The echo experiments drop lower than the actual edge modes, suggesting there is miscalibration in the experiment or some other coherence errors. I therefore find it difficult to accept this reference experiment as evidence that the observed decay is due to only decoherence.

Authors’ response: We thank the Referee for catching this subtle point, since the echo dynamics should decay more slowly serving as the upper bound of the edge-mode dynamics if only depolarizing noise exists. We have rerun this echo experiment using nominally the same parameters after re-calibrating the experimental system, and obtained the correct picture as shown in Figure R8.

We do not know what exactly happened during the previous echo experiment. It is possible that some qubits were interfered by unexpected two-level system (TLS) defects that are mobile in the frequency domain [Fluctuating TLS defects have been confirmed in a number of experiments including, e.g., Phys. Rev. Lett. 121, 090502 (2018)]. Consequently, the gate parameters could drift slightly yielding extra coherent error during the previous echo dynamics.

In the revised manuscript, we have incorporated the new data in Figure R8 into Fig. 2b and Extended Data Fig. 3a of the main text and the above-mentioned discussion in Supplementary Section 2G.

Comment 8 of Referee #3:

3. Fig. 4: Why reduce the system size to just 8 qubits in this case? I understand 100-qubit spectroscopy is difficult, but it seems that the authors should at least attempt to measure the spectra for a few system sizes up to ~ 20 and see that the gap δ responsible for prethermal protection does not disappear at large system sizes.

Authors’ response: We thank the Referee for this valuable suggestion. In the original manuscript, we performed spectroscopy on the 8-qubit system to clearly display all relevant gaps (Δ_e , Δ_o , ζ_o , ζ_e , and δ) and

their dependence on J_o/J_e within a single figure.

We agree with the Referee that including spectroscopic measurements for larger system sizes and analyzing the relationships between various gaps and system size would significantly strengthen our conclusion regarding prethermal protection. Yet, we remark that conducting spectroscopy for our model at larger system sizes presents substantial experimental challenges: Each Trotter step comprises 4 layers of two-qubit gates and 2 layers of single-qubit gates, which is doubled compared to the related work in Ref. [39]. This results in deeper circuits and an effectively shorter lifetime for the operators $\tilde{Z}_{L,R}$, making it difficult to extract all eigenmodes from $\tilde{Z}_{L,R}(\omega)$ in large systems.

To mitigate this challenge, we enhanced the spectroscopy method: Instead of merely measuring $\tilde{Z}_{L,R}(t)$, we now also measure the dynamics of a set of operators:

$$O_{L,i} = \left(\prod_{k=1}^{2i} \sigma_k^x \right) \sigma_{2i+1}^z, \quad O_{R,i} = \left(\prod_{k=1}^{2i} \sigma_{2N+1-k}^x \right) \sigma_{2(N-i)}^z, \quad (3)$$

which are exactly the bulk terms in the PSZMs when H_1 only contains single-qubit σ_i^x perturbations. Under the Jordan-Wigner transformation, $O_{L,i}, O_{R,i}$ are mapped to $c_{2i+1} + c_{2i+1}^\dagger, c_{2(N-i)} + c_{2(N-i)}^\dagger$, respectively, in the bulk of the upper and lower Kitaev chains. These operators exhibit larger overlap with bulk eigenmodes. This enhanced approach enables us to perform spectroscopy up to a system with $N = 16$ qubits, doubling the system size examined in the original manuscript ($N = 8$). We measured the dynamics of $O_{L,i}(t), O_{R,i}(t)$ up to $i = 3$ (which are 7-body operators) and present their Fourier transformed results for

Figure R8: Remeasured results of edge operators for the case of $\{|\Psi_0\rangle\}$ and echo experiments in Fig. 2b and Extended Data Fig. 3a of the main text after we returned up the 100-qubit system and carefully recalibrated all the gates. Error bars represent the standard deviation over five rounds of measurements, with each taking 10,000 shots.

the 16-qubit system in Figure R9. From this figure, we find that each $O_i(\omega)$ characterizes specific bulk eigenmodes more effectively than $\bar{Z}(\omega)$, and their average provides a clearer overall spectrum. We display the spectra measured for system sizes ranging from $N = 8$ to $N = 16$ in Figure R10, confirming that the gap δ persists and does not close with increasing system size.

In this revised manuscript, we have replaced Fig. 4 in the main text with a new figure plotting spectroscopic results for the 16-qubit system. Spectroscopic results for different system sizes are integrated into a new Extended Data Fig. 6, and the improved spectroscopy method is described in detail in Supplementary Information Sec. 1G.

Comment 9 of Referee #3:

4. I'm a little confused how it could be that X_L and Z_L can simultaneously start off at 1 (in e.g. Fig. 2b). Shouldn't they form a vector of length at most unity on the Bloch sphere?

Figure R9: Experimental results of enhanced spectroscopy for the 16-qubit system. The left (right) column shows the spectra of operators corresponding to the upper (lower) Kitaev chain. The last row shows the averaged spectra over $\{Z(\omega), O_{1,2,3}(\omega)\}$.

Figure R10: Spectroscopy for different system sizes. The results for system sizes ranging from 8 to 16 are presented from top to bottom. The first two columns show the averaged spectra of the upper and lower Kitaev chains as functions of ω and J_o/J_e . The third column shows their combination as the spectra of the entire system. The presented spectra are obtained by averaging the Fourier spectra of different operators $O_{L,i}$, where the averaged sets are chosen to be $i \in \{0\}$ (which are \tilde{Z}_L, \tilde{Z}_R) for $N = 8$, $i \in \{0, 1, 2\}$ for $N = 10, 12$ and $i \in \{0, 1, 2, 3\}$ for $N = 14, 16$.

Authors' response: We thank the Referee for raising this point. Indeed, when starting from the same initial state, the sum of the norms of \tilde{X}_L and \tilde{Z}_L cannot exceed 1. However, in our experiments, we measure $\{\tilde{X}_L, \tilde{X}_R\}$ and $\{\tilde{Z}_L, \tilde{Z}_R\}$ by preparing distinct initial states, each aligned with the respective edge mode direction. For the zero-temperature manifold $\{|\Psi_0\rangle\}$, they are $|\tilde{+}\rangle_L |\tilde{+}\rangle_R, |\tilde{0}\rangle_L |\tilde{0}\rangle_R$, as defined in Eqs. (3-4) in the Methods section. In the revised manuscript, we have explicitly clarified that \tilde{X}_L and \tilde{Z}_L were measured using different initial states to avoid possible confusions.

In summary, we greatly appreciate the Referee's meticulous review and insightful and constructive com-

ments/suggestions, which have guided us to substantially improve the manuscript. Based on their report, we have spent great efforts in conducting additional experiments, including measurements of edge mode spatial profiles, comparison of logical and physical qubit resilience to noises, and spectroscopy for larger system sizes using an improved method. These new results not only underscore the many-body nature of the PSZMs but also highlight their potential as noise-resilient carriers of quantum information at finite temperature. We hope this significantly improved version will satisfy the Referee and convince them to recommend publication of this work in Nature.

Response to Referee #1

We sincerely thank the Referee for their careful reading of the revised manuscript and greatly appreciate their valuable suggestions, which have led us to further clarify several important points. The detailed responses to the Referee's comments are provided below.

Comment 1 of Referee #1:

I am pleased to see the authors have taken my suggestions onboard, most notably including Matrix Product State simulations of the 100 qubit system in the appendix.

It is curious, however, that these are not mentioned in the main text. It appears from the Supplemental that these classical simulations are able to resolve the dynamics of the system at hand (on the timescales considered in the experiment – 100 qubits, up to 30 cycles) to high numerical accuracy (certainly much higher than the processor). In fact the bond dimensions required mean a laptop could be used for this purpose. I think it would be important to make this clear to readers and even include such classical data in the main plots as it is essentially exact simulation data.

Authors' response: We thank the Referee for this important comment regarding the classical simulations, which helps us further clarify the importance of our experiment. As the Referee correctly pointed out, the MPS simulations are exact and do not account for experimental noises. We agree that these exact simulations provide a highly accurate description of the ideal, noise-free system dynamics. However, a crucial finding from our experiment is the robustness and longevity of the edge modes as prethermal strong zero modes, even when subjected to experimental noises. This crucial aspect cannot be captured by the exact MPS simulations, and we believe that performing numerical simulations of our 100-qubit system up to 30 Trotter cycles, taking experimental noises into account, would be significantly more challenging and is beyond our current capacity. Therefore, our experimental demonstration of this noise robustness is essential, highlighting the practical implications and significance of these robust edge modes.

In the revised manuscript, we have mentioned in the main text about the exact MPS simulations, and have clarified the difficulty of numerical simulations of our experiment with noises.

Comment 2 of Referee #1:

This, and the other referees comments about the fact the physics of the system can be observed without the need for 100 qubits, leads me to now have some concern about the suitability manuscript for Nature. Whilst I find it impressive the authors are able to realize these topological edge modes in a many-body system, the setup does not gain us physical insight beyond what could have been achieved via simpler, more efficient means.

Authors' response: We thank the Referee for this important question about the necessity of the large system size used in our experiment. We emphasize that while some aspects of edge modes can be explored in smaller systems, the large system size is essential for uncovering and systematically studying several novel phenomena that are either inaccessible or extremely challenging to observe in shorter chains.

First, beyond just edge behavior, a key aspect of our experiment is to demonstrate the emergent $U(1)\times U(1)$ symmetry in the prethermal regime due to dimerization, where excitations deep in the bulk play a crucial role. As illustrated in Fig. 3 of the main text, a large system enables us to explore the rich bulk excitation

dynamics and the distinct interacting behaviors among propagating bulk excitations, which are difficult to observe in short chains. In addition, our work identifies two different prethermal mechanisms. Their coexistence in a large system over a broad energy range is not *a priori* expected or previously observed. Therefore, the large system size in our experiments is essential for systematically studying these interesting phenomena.

Second, scaling to large system sizes is a promising route for discovering and verifying new physics. As shown in the recent theoretical work [Phys. Rev. B 111, L201114 (2025)], strong zero modes are not limited to the ends of chains but can also emerge at interfaces between distinct phases. Our extensive bulk allows us to engineer different phases within separate regions of the chain, leading to multiple boundary modes occurring at these interfaces. This approach gets rid of the geometric constraints on performing logical operations using only edge modes and offers a scalable way to increase the number of strong zero modes within a single 1D chain. This opens up exciting possibilities for quantum information processing that are only accessible with large system sizes.

Third, this experiment serves as an alternative benchmark of our newly developed 125-qubit superconducting processor, showcasing its high programmability and low-error parallel gates. The advanced capabilities of our platform, particularly its programmability and low error rates, enable the exploration of extended models exhibiting similar prethermal mechanisms. For instance, the model studied in our experiment can be readily extended to two dimensions to explore higher-order SPT phases and corner modes, which our device can implement with a considerably large system size. This demonstrates the utility of our platform as a powerful tool for future quantum simulation research.

Comment 3 of Referee #1:

Thus the main result here is the devices ability to physically realize these edge modes and their potential for quantum information storage. In my opinion, without evidence i) the device has been used to discover something unknown, ii) simulate a system outside the reach of other methods, or iii) a clearer case for the utility of these realized topological edge modes, I think the paper is not suited for Nature.

Authors' response: We thank the Referee for these inspiring comments. We agree that realizing topological edge modes and demonstrating their potential for quantum information storage are important aspects of our findings. However, we respectfully disagree that our setup does not gain physical insight beyond what could have been achieved by simpler means, or that it lacks clear utility or novel discovery.

Regarding i), our work represents the first experimental observation and characterization of finite-temperature prethermal strong zero modes in a disorder-free, interacting system. This is distinct from previous experimental works (e.g., Refs. [37-39]), which all incorporate disorders into their models. A defining feature of our disorder-free, interacting model is that, unlike the nearly localized excitations typically found in disordered systems, our system exhibits dynamic bulk excitations. This allowed us to experimentally investigate their propagation and interaction with edge modes, providing previously inaccessible insights. Furthermore, we present the first experimental observation of resonance phenomena in prethermal strong zero modes. This behavior arises from the emergent $U(1)\times U(1)$ symmetry in the prethermal regime, a feature that is also absent in prior experimental studies.

Regarding ii), we would like to first emphasize that the aim of our experiment is not to demonstrate quantum advantage beyond classical computations. Instead, the primary focus of our experiments is to investigate the prethermal strong zero modes and excitation dynamics under realistic conditions, including experimental

noises. As discussed in our previous response, noisy simulation of our 100-qubit system up to 30 Trotter cycles to capture the robustness of edge modes against experimental noises could be challenging for classical computational methods.

Regarding iii), we demonstrate a direct utilization of these prethermal strong zero modes for quantum information processing: We experimentally use these robust, long-lived edge modes to encode a logical Bell state and protect it against bulk excitations. In addition, we demonstrate that this encoding provides enhanced robustness against symmetry-preserving noises compared to encoding the same state on physical qubits. This experimental demonstration of improved resilience to excitations and noises, inherent to the prethermalization and topological protection, provides a clear and compelling case for their utilization in building more robust quantum information storage at finite temperature.

Comment 4 of Referee #1 (Remarks on code availability):

Everything is appropriate. I appreciate the addition of a README.

Authors' response: We thank the Referee for taking the time to verify our code and data.

In summary, we greatly appreciate the Referee's careful reading of our revised manuscript and their valuable suggestions. Following these suggestions, we have better clarified our findings and contributions: We observed robust edge modes against experimental noises and studied the bulk-edge interactions in a disorder-free system, along with demonstrating the practical utilization of these edge modes in quantum information processing tasks. We made revisions to the manuscript accordingly and improved the presentation.

Response to Referee #3

We sincerely thank the Referee for their careful reading of the revised manuscript and greatly appreciate their constructive suggestions and kind recommendations. The detailed response to the Referee's comments is provided below.

Comment 1 of Referee #3:

I have read through the revised manuscript and the authors' response to my questions. The updated manuscript is significantly improved over the initial version, with the new data substantiating the authors' claims much better. I therefore recommend the work for publication.

Authors' response: We thank the Referee for judging that our work "is significantly improved over the initial version," and that "the new data substantiating the authors' claims much better." We greatly appreciate their recommendation for publishing our work in Nature.

Comment 2 of Referee #3:

p.s. I still share my initial concern that the edge modes are highly localized quantities and unaffected by majority of the qubit chain, a suspicion confirmed by the authors' new edge mode profile measurements showing the weights are concentrated in just the outermost qubit. The system (away from the critical point) will therefore never be complex enough to require a quantum computer to investigate.

Authors' response: We thank the Referee for this comment. We agree that the edge modes are highly localized when system parameters are dimerized and far from resonances. However, our experiment explores conditions beyond this, specifically when system parameters are near resonant points. In these cases, we observed distinct phenomena such as a sharply reduced edge mode lifetime and sign flips in their spatial profiles. These observations suggest more complex interactions between edge modes and bulk excitations, leading to a much more extended edge profile.

We fully agree with the Referee that further studies of the system behavior near the resonant points are quite important and represent a promising direction for future work: Theoretically, it is interesting and important to figure out the exact form of prethermal strong zero modes in higher orders. Experimentally, further improvement of gate fidelities so as to implement deeper circuits and probe longer-time dynamics is demanded, enabling one to measure more extended spatial profiles and better characterize the edge mode behavior. It would also be interesting to explore these complex physical phenomena with corner modes in two-dimensional systems, a capability our programmable 125-qubit processor is well-suited to investigate. These future directions will leverage our large-scale quantum platform to uncover richer physics beyond the reach of classical simulations and smaller quantum systems.